# A protein sequence-based deep transfer learning framework for identifying human proteome-wide deubiquitinase-substrate interactions

Yuan Liu[1,5], Dianke Li [1,2,5], Xin Zhang [1,5], Simin Xia [1,3], Yingjie Qu[1], Xinping Ling [1,4], Yang Li[1], Xiangren Kong[1], Lingqiang Zhang[1], Chun-Ping Cui[1] ✉ & Dong Li[1] ✉

Protein ubiquitination regulates a wide range of cellular processes. The degree of protein ubiquitination is determined by the delicate balance between ubiquitin ligase (E3)-mediated ubiquitination and deubiquitinase (DUB)-mediated deubiquitination. In comparison to the E3-substrate interactions, the DUB-substrate interactions (DSIs) remain insufficiently investigated. To address this challenge, we introduce a protein sequence-based ab initio method, TransDSI, which transfers proteome-scale evolutionary information to predict unknown DSIs despite inadequate training datasets. An explainable module is integrated to suggest the critical protein regions for DSIs while predicting DSIs. TransDSI outperforms multiple machine learning strategies against both cross-validation and independent test. Two predicted DUBs (USP11 and USP20) for FOXP3 are validated by "wet lab" experiments, along with two predicted substrates (AR and p53) for USP22. TransDSI provides new functional perspective on proteins by identifying regulatory DSIs, and offers clues for potential tumor drug target discovery and precision drug application.

Ubiquitin, a 76-amino acid protein that is widely expressed and highly conserved in eukaryotes, is conjugated to substrate proteins through a tightly regulated cascade involving the ubiquitin activating enzyme (E1), the ubiquitin conjugating enzyme (E2), and the ubiquitin ligase (E3), and removed from the substrate by deubiquitinase (DUB)[1]. Protein ubiquitination is a highly prevalent post-translational modification (PTM) that regulates a wide range of cellular processes, including cell proliferation, survival, differentiation and cellular signal transduction[2]. Like most PTMs, ubiquitination is a dynamic and reversible process[3]. The degree of protein ubiquitination is determined by the delicate balance between specific E3-mediated ubiquitination and DUB-mediated deubiquitination[4]. Both E3s and DUBs have been elegantly leveraged for drug development in the forms of PROTAC[5] (Proteolysis-targeting chimeras) and DUBTAC[6] (Deubiquitinase-targeting chimeras for targeted protein stabilization) technology. Compared to E3s, the less-studied DUBs have been found to exert distinct functions such as oncogenic, tumor-suppressive or context-dependent roles in tumorigenesis, mainly by affecting the protein stability, enzymatic activity or subcellular localization of their substrates[7,8]. Studies of DSIs have shed light on the mechanisms of cancer therapy and may provide new avenues for drug design.

[1]State Key Laboratory of Medical Proteomics, Beijing Proteome Research Center, National Center for Protein Sciences (Beijing), Beijing Institute of Lifeomics, Beijing 102206, China. [2]State Key Laboratory of Farm Animal Biotech Breeding, College of Biological Sciences, China Agricultural University, Beijing 100193, China. [3]School of Basic Medical Sciences, Anhui Medical University, Hefei 230032, China. [4]College of Life Sciences, Hebei University, Baoding 071002, China. [5]These authors contributed equally: Yuan Liu, Dianke Li, Xin Zhang. ✉e-mail: cui_chunping2000@aliyun.com; lidong.bprc@foxmail.com

Several experimental methods have been developed for identifying DSIs, such as protein microarrays[9], global protein stability profiling[10], mass spectrometry[11] and live phage display library[12]. However, due to the substrates' low expression levels and their intrinsically weak interactions with enzymes, these methods are often laborious, time-intensive, expensive and inefficient. As a result, although there are >100,000 ubiquitin sites on over 9,000 proteins in the ubiquitination site resource Ubisite[13], only <900 human DUB-substrate relationships are collected in corresponding database[14], which means that only a small proportion of ubiquitinated proteins have the known corresponding DUB information. Therefore, it is an urgent need to identify proteome-wide DSIs through bioinformatics strategy.

In 2022, we proposed UbiBrowser 2.0 (UB2), a computational algorithm based on Naïve Bayesian classifier to predict human DSIs by combining multiple types of heterogeneous biological features, including homology, enriched protein domain and function, protein-protein interaction (PPI) network topology, and inferred DUB recognition consensus motif[14]. UbiBrowser 2.0 is a popular publicly available bioinformatics tool capable of proteome-wide DSIs prediction. However, this method relies on feature engineering and its application on proteome scale is inevitably hindered by the lack of training datasets. To address these challenges, we introduce TransDSI, an explainable transfer learning architecture based on protein sequence only. TransDSI is pre-trained by sequence similarity network between 20,398 proteins, and fine-tuned by 863 experimentally validated DSIs. Meanwhile, TransDSI presents an explainable module that can suggest the critical protein regions for DSIs while predicting DSIs, partially capturing the protein structural basis of DSIs. We conducted proteome-wide scanning and generated a predicted DUB-substrate interaction dataset (PDSID). Two predicted DUBs (USP11 and USP20) for FOXP3, along with two predicted substrates (Androgen receptor AR and Cellular tumor antigen p53) for USP22 were validated by our "wet lab" experiments, contributing to tumor immune escape-related drug target discovery and precise application of anti-tumor agent, respectively. TransDSI also provides a new perspective for disease omics data analysis by identifying regulatory DUBs for significantly dysregulated proteins in hepatocellular carcinoma (HCC). To facilitate the usage of TransDSI, we made the PDSID and the corresponding program codes available on github (https://github.com/LiDlab/TransDSI).

## Results

### An overview of TransDSI

Deep learning framework of TransDSI consists of four modules. Firstly, given the primary structures of human proteins, the protein coding module utilizes conjoint triad (CT) method[15] to generate protein sequence features and BLAST[16] to construct a sequence similarity network (SSN) (Fig. 1a). Next, the self-supervised learning module exploits a variational graph autoencoder (VGAE)[17] to process them simultaneously and pre-train a Graph Convolutional Network (GCN)[18] encoder which can effectively compress complex graph structure data in non-Euclidean space into low-dimensional numerical vectors while preserving as much relevant information from the original input as possible (Fig. 1b, more details can be found in Methods). After that, DSI-Predictor module adopts transfer learning mechanisms to initialize parameters from pre-trained GCN encoder and performs fine-tuning using DUBs and their corresponding substrates that are involved in gold standard dataset. Our gold standard positive dataset (GSP) was obtained by manual curation by experts, and each pair of interaction is supported by traceable literature evidence, while the gold standard negative dataset (GSN) was obtained by randomly sampling protein-protein interactions from the complement graph of the GSP while preserving its network topology (Details in Supplementary Fig. 1 and Methods Section). Specifically, DSI-Predictor concatenates embeddings of DUBs and substrates and uses a multilayer perceptron (MLP)[19] with four fully connected layers to predict whether

there exists a functional interaction between DUB and substrate (Fig. 1c). Finally, we developed PairExplainer, an explainable module to obtain an optimized mask highlighting the contribution of different positions in the protein sequence to prediction, which might partially explain the protein structural basis of DSI (Fig. 1d).

A key insight of our framework is that there might be many evolutionarily conserved protein regions in the proteome that contribute to DSIs. TransDSI aims to extract these conserved regions from proteome-scale, and exploit evolutionary information of known DSIs to predict unseen DSIs. Intuition tells us that these conserved regions are likely to contribute the most to the predicted results and can be further investigated using perturbation-based explainable methods. More details of the TransDSI can be found in Methods.

### TransDSI has the ability to predict true DSIs

To test the effectiveness of the TransDSI framework, we initially conducted a comparative analysis with UbiBrowser 2.0[14] developed by our group, which is a popular publicly available bioinformatics tool that can predict proteome-wide DSI. Meanwhile, we constructed five additional DSI prediction systems employing several machine learning methods including random forest (RF), support vector machine (SVM), eXtreme gradient boosting (XGBoost), logistic regression (LR), and K-nearest neighbors (KNN) based on the same training dataset and protein sequence features as that of TransDSI. Additionally, to demonstrate the importance of using sequence features only, we also established a variant for UB2: UB2 without PPI network topology and Gene Ontology (GO) term pair features (UB2 w/o NT and GO), where we removed PPI network topology and GO term pair features from UB2, and only sequence features (domain pair and recognition consensus motif) are used.

We compared the performance of TransDSI, UB2, and other machine learning methods for DSI prediction on two distinct subsets within the GSP (Supplementary Data 1): (1) A 5-fold cross-validation dataset collected from a manual curation of PubMed prior to June 2018; (2) An independent test set derived from literature published between June 2018 and August 2021, which does not intersect with the 5-fold cross-validation dataset.

The performance of TransDSI was evaluated using both area under the receiver operating characteristic curve (AUROC) and that under the precision recall curve (AUPRC). AUROC is a robust metric for evaluating model discrimination between positive and negative examples; while AUPRC excels in identifying true positives especially in imbalanced datasets[20]. The more a test's AUROC/AUPRC approximates to 1.0, the higher its overall efficacy will be. TransDSI achieved an AUROC of 0.83 (95% CI = 0.79–0.87) and AUPRC of 0.95 (95% CI = 0.92-0.96) in 5-fold cross-validation, and an AUROC of 0.75 (95% CI = 0.71–0.80) and AUPRC of 0.77 (95% CI = 0.70-0.82) in independent test, which outperforms all other methods (Fig. 2a–d). Notably, the performance of UB2 dropped dramatically after removing features such as homology DUB-substrate interaction, GO term pair, and PPI network topology (Fig. 2a), which suggests that UB2 cannot achieve ideal prediction performance without prior knowledge of the two proteins involved in DSI (Fig. 2b). This emphasizes the importance of using protein sequence features only for prediction in TransDSI, since protein sequences are the easiest to obtain. In real-world applications, higher specificity is desired to improve the success rate of experimental validation. In independent test, TransDSI has higher coverage in scenarios requiring high specificity compared to UB2. For example, when specificity is set to 0.90, TransDSI has a recall of 45.0 %, which is much higher than that of UB2 (21.1%). This result suggests that, at this particular threshold, TransDSI possesses a more robust capacity to predict positive DSIs and delivers enhanced performance. Conversely, UB2 compromises recall in pursuit of elevated accuracy, consequently yielding reduced coverage.

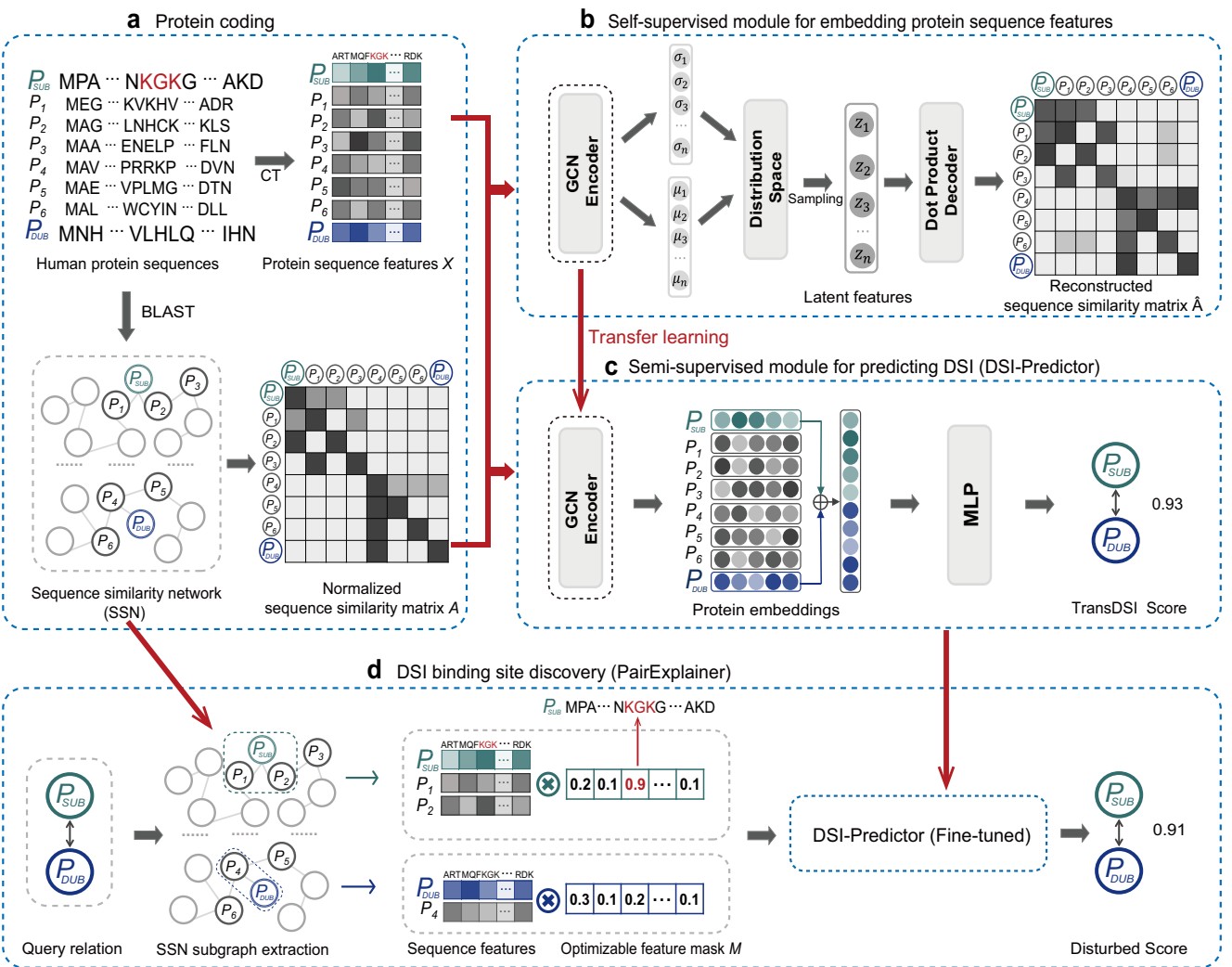

**Fig. 1 | Framework of TransDSI.** The model comprises of four components.
**a** Protein coding: This module takes the primary structures of human proteins including DUBs ($P_{DUB}$), substrates ($P_{SUB}$) and other proteins ($P_1$-$P_6$) as input and generates a variety of features for use in downstream deep learning modules. The amino acid sequences of proteins are encoded using CT method[15] and serve as protein sequence features $X$ (darker colors indicate higher conjoint triad frequency). Additionally, a sequence similarity network (SSN) was created using BLAST and transformed into a normalized sequence similarity matrix $A$ (darker colors indicate higher similarity). **b** Self-supervised module for embedding protein sequence features: This module employs a VGAE consisting of a GCN encoder and a dot product decoder, which is used to generate a pre-trained encoder based on the evolutionary information from the SSN and the protein sequence features (see Methods for details of $\sigma$, $\mu$, $z$ and $\hat{A}$). **c** Semi-supervised module for predicting DSI

(DSI-Predictor): The GCN encoder of DSI-Predictor is initialized using the parameters transferred from the self-supervised module to produce the protein embeddings. The embeddings of DUBs and their corresponding substrates are concatenated and utilized for fine-tuning the semi-supervised module. The final prediction score ("TransDSI Score") for each candidate DSI is obtained by feeding the concatenated embeddings to an MLP. **d** Explainable module for DSI binding site discovery (PairExplainer): Firstly, we froze the parameters of the fine-tuned DSI-Predictor module. Then, based on the query relation ("$P_{DUB}$-$P_{SUB}$"), the relevant subgraphs of the DUB and substrate are extracted and a Hadamard product operation is performed between the features of the nodes in each subgraph and corresponding optimizable feature mask $M$. In the optimized mask, certain amino acid sequence features such as "KGK" are highlighted, indicating that it may suggest the structure basis for the interaction between DUBs and substrates.

In addition, F1-score, positive predictive value (PPV), and negative predictive value (NPV) were also employed as evaluation metrics. AUROC and AUPRC are comprehensive metrics that consider all possible thresholds, while sensitivity, specificity, PPV, NPV, and F1-score are metrics for optimal specific threshold which was identified by the Youden Index[20,21]. Across all these metrics, TransDSI consistently demonstrates superior predictive performance compared to other prediction methods (Supplementary Data 2).

To further test the robustness and efficacy of the proposed TransDSI model to handle real-world scenarios, we performed 30 iterations of random sampling for the negative set construction (protein-protein interaction data). The AUROC and AUPRC of TransDSI on the independent test set exhibited remarkable stability, with standard deviations of only 0.017 and 0.025, respectively (Supplementary

Data 2). Meanwhile, we examined the performance of TransDSI and five machine learning methods across diverse negative/positive ratios (1:1, 2:1, 5:1, and 10:1). TransDSI outperforms other methods against all ratios (Supplementary Data 3). These results demonstrate that TransDSI has satisfactory robustness and the potential for practical applications.

TransDSI was then used to perform a large-scale proteome-wide DSI scanning, resulting in a predicted DUB-substrate interaction dataset (PDSID) with 19,461 predicted interactions between 85 DUBs and 5,151 substrates (Supplementary Data 4). This predicted DUB-substrate interaction network presents a scale-free degree distribution (linear model fitting R2 index = 0.93)[22].

In addition, Gene Ontology (GO)[23,24] enrichment analysis revealed that DUBs and their known substrates, as well as predicted

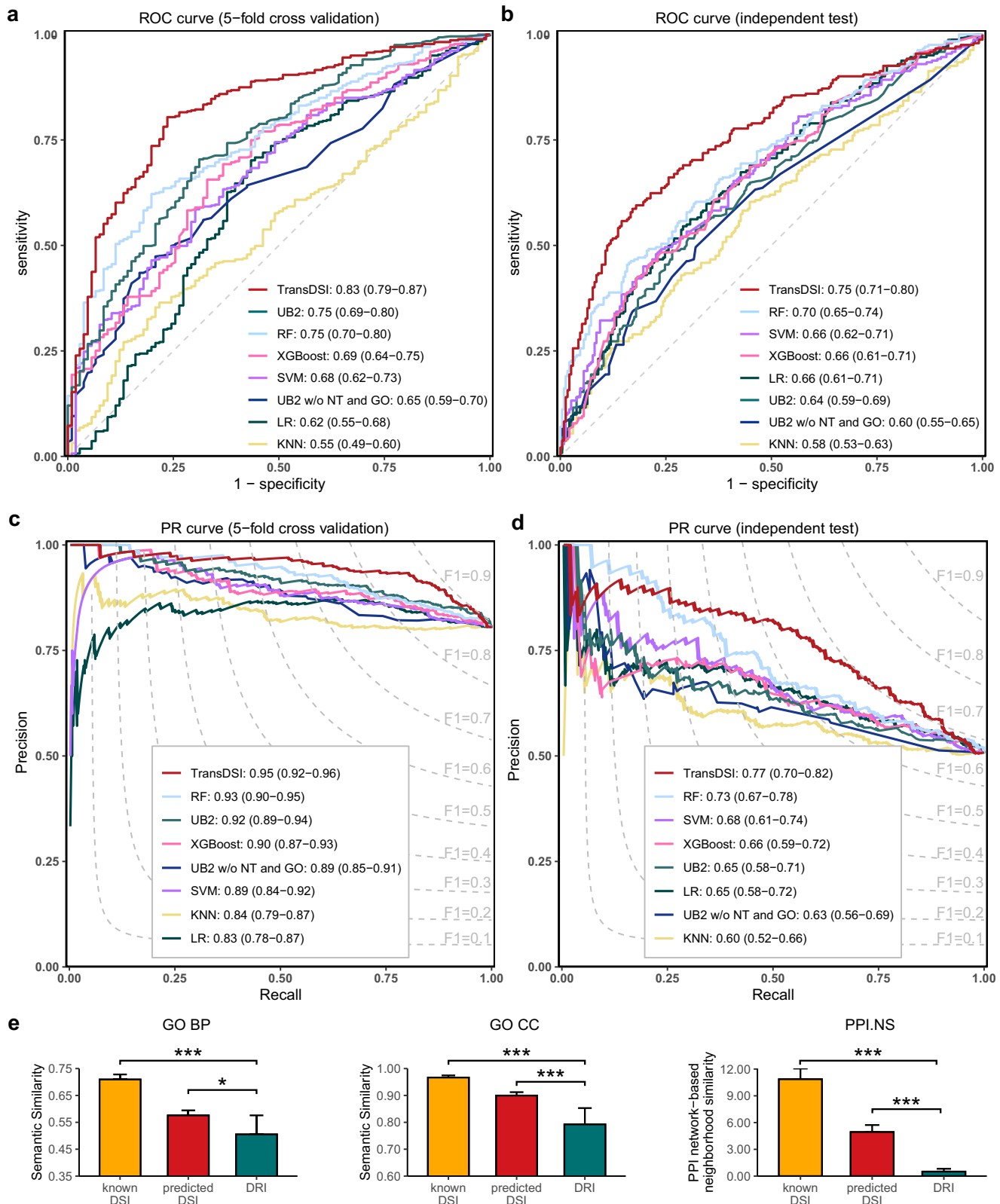

substrates, tend to be associated with similar functional categories (Supplementary Data 5), such as protein deubiquitination ($ER_{DUB} = 61.11$; $ER_{known\ SUB} = 9.80$; $ER_{predicted\ SUB} = 9.80$), non-recombinational repair ($ER_{DUB} = 8.64$; $ER_{known\ SUB} = 6.76$; $ER_{predicted\ SUB} = 3.80$), and regulation of cytokine-mediated signaling pathways ($ER_{DUB} = 7.36$; $ER_{known\ SUB} = 5.76$; $ER_{predicted\ SUB} = 3.46$).

We further tested GO term similarity between DUBs and their predicted substrates in terms of biological process (BP), cellular component (CC). As Fig. 2e showed, consistent with known DSIs, DUBs and their predicted substrates tend to be involved in the same biological process, locate in the same cellular component compared to randomly sampled DUB-random protein interactions (DRIs). Besides,

**Fig. 2 | Performance evaluation of TransDSI. a–d** Performance of various models (TransDSI, UB2, and five machine learning models) for the prediction of DSIs, evaluated through 5-fold cross-validation and independent test, respectively. Five machine learning models are random forest (RF), support vector machine (SVM), eXtreme gradient boosting (XGBoost), logistic regression (LR), and K-nearest neighbors (KNN). The ROC curves of the assessment models demonstrate sensitivity and specificity (**a, b**) and the PR curves of the assessment models precision and recall (**c, d**) against a particular prediction score cutoff, with each point on the curves representing the respective values. The 95% confidence intervals (95% CIs) of the sensitivity (precision) at the given specificity (recall) points are computed. The reference line indicates a non-informative prediction with an AUROC of 0.5

(**a, b**) or a prediction with a constant F1 score across different thresholds (**c, d**). **e** Multidimensional association features for known DSIs (orange bars, n = 863 DSIs), predicted DSIs (red bars, n = 1000 DSIs), and randomly screened DRIs (green bars, n = 1000 PPIs). The feature scores include GO term similarity in terms of biological process (GO BP), cellular component (GO CC) and PPI network-based neighborhood similarity (PPI.NS). The colored bars represent the average value of various association features, with the error bars marking 95% confidence intervals. The One-tailed Wilcoxon test is used to test the difference between DRIs and known or predicted DSIs (*P-value < 0.05, ***P-value < 0.001). The exact P-values are: GO BP (1.9e-8, 2.2e-2); GO CC (4.4e-22, 1.3e-5); PPI.NS (9.5e-205, 1.3e-297). DRI, DUB-random protein interaction. Source data are provided as a Source Data file.

we found DUBs and their predicted substrates tend to locate in tightly connected subgraphs within PPI network. These findings are consistent with the previous reports[25] and imply the reliability of the PDSID.

Motivated by the abundant and balanced dataset of E3-substrate interactions (ESIs) available in our UbiBrowser 2.0[14] (containing 4,068 ESIs), we further investigated the applicability of the TransDSI deep learning framework for predicting ESIs (Supplementary Fig. 2). Interestingly, on an independent ESI test set, this deep learning framework outperforms other machine learning systems (Supplementary Fig. 3), which indicates that the TransDSI deep learning framework has certain generalization ability.

### Explainable module of TransDSI provides partial insights into the protein structural basis of DSI

Some DUB-substrate interactions are mediated by the interacting protein domains and motifs[26]. We developed a perturbation-based explainable module PairExplainer, which allows us to identify critical protein sequence features.

Firstly, we froze the parameters of the fine-tuned DSI-Predictor module. Then, based on the query DSI, the relevant protein sequence features involved in subgraphs of the DUB and substrate were extracted and disturbed with optimizable feature masks. These masks were trained by minimizing the discrepancy between the disturbed prediction score ("Disturbed Score") and the original one ("TransDSI Score"). The optimized mask highlights the contribution of different positions in the protein sequence to prediction (Fig. 1d).

Based on a given query DSI, the algorithm of PairExplainer is equivalent as using a sliding window of 3 residues in length to move along the protein sequence with a step size of 1 residue, performing in silico "knockout" by removing all residues except for the triad within the sliding window. It observes the impact on "TransDSI Score" after knockout and assigns importance scores to each residue based on the magnitude of the effect within all three triads that the residue is involved in (Fig. 3a).

We predicted the residue-level importance scores for DUBs and their substrates in the GSP and provided the top 10 residues with the highest contribution to the interaction for each protein (Supplementary Data 6). We collected all experimentally confirmed DSI binding sites in literature (9 sites, involving 1 DUB and 5 substrates) and used them to assess these identified residues (Supplementary Data 7). Interestingly, some of these protein features can partially explain the structural basis of DSI. For example, PairExplainer successfully captured the KxxxKxK motif on DNA (cytosine-5)-methyltransferase 1 (DNMT1) and Ubiquitin-like PHD and RING finger domain-containing protein 1 (UHRF1), which is known to bind to Ubiquitin carboxyl-terminal hydrolase 7 (USP7) and is one of only two known USP7 recognition motifs (Supplementary Data 7).

To further elucidate this finding, we selected a real crystallographic structure of the USP7 and DNMT1 complex as an example[27]. Figure 3b illustrates the importance of each residue on the DNMT1 sequence for the interaction. A concentration of red, indicating high importance, can be observed on the lysine residues (K1111/

K1113/K1115) in the KG repeat zone (residues 1,109–1,119) of DNMT1. We presented surface representations of the projection of the heatmap on major interfaces of the DNMT1–USP7 complex (determined by Cheng et al.[27]. using X-ray crystallography experiments), highlighting the critical residues involved in the interactions through stick representation (Fig. 3c). These findings are in line with previous experimental studies showing that the interaction between USP7 and substrate DNMT1 is primarily mediated by the acidic pocket of USP7 and the lysine residues in the KG repeat zone of DNMT1[27].

Furthermore, PairExplainer can also identify the MATH structural domain mediating the competitive binding of USP7 to p53 and MDM2 (Supplementary Data 7). This domain was reported to play a crucial role in the regulation of the p53-MDM2 signaling pathway, which has implications for understanding the mechanism of tumor suppression and the development of new anticancer drugs[28].

### Experimental validation of predicted DSIs and their application in oncology research

To verify whether TransDSI can accurately predict potential DSIs, we selected several DSIs with certain biological significance and high ranking for experimental validation. A series of biochemical experiments were conducted to verify these interactions and the regulatory effect of DUBs on ubiquitination of substrates. These are promising to expand the understanding of tumor development mechanisms from the perspective of deubiquitination-regulated protein homeostasis, contributing to the refinement of cancer patient stratification and the development of personalized treatment strategies.

Potential DUBs that regulate FOXP3. Forkhead box protein P3 (FOXP3), a major transcription factor, mediates the suppressant effect of regulatory T (Treg) cells on antitumor immune responses[29]. The induction of FOXP3 transcription is the result of synergy between TGF-β receptor (TGFβR) activated SMAD3/4 and T cell receptor (TCR) activated Nuclear factor of activated T-cells (NFAT, Fig. 4a)[30]. Elevated levels of FOXP3 in multiple tumor types has been reported to be associated with worse overall survival[31–33]. However, due to the critical role of FOXP3 in regulating autoimmunity, it cannot be directly targeted for therapy[34]. Identification of DUBs that regulate both FOXP3 and its upstream regulators will provide valuable insights for the development of potential therapeutic targets related to FOXP3 regulation.

Utilizing TransDSI, we have predicted seven potential DUBs for FOXP3. We selected five DSIs with relatively high ranking for validation (USP18, score: 0.955; UCHL1, score: 0.954; UCHL3, score: 0.954; USP11, score: 0.947; USP20, score: 0.946). We observed that both USP11 and USP20 can act as DUBs to deubiquitinate FOXP3 (Fig. 4a). USP11 has been reported to be able to enhance TGF-β-mediated TH17 cell differentiation and stabilize FOXP3 expression, thereby maintaining the suppressive capacity of Tregs[35]. However, the underlying molecular mechanism is still unclear. We found that USP11 can bind FOXP3 directly and remove the ubiquitin conjugation on FOXP3, enhancing the stability of FOXP3 (Fig. 4b, c). Considering tumor-infiltrating FOXP3+ Treg cells may promote the immune escape of cancer cells, our findings suggest that USP11

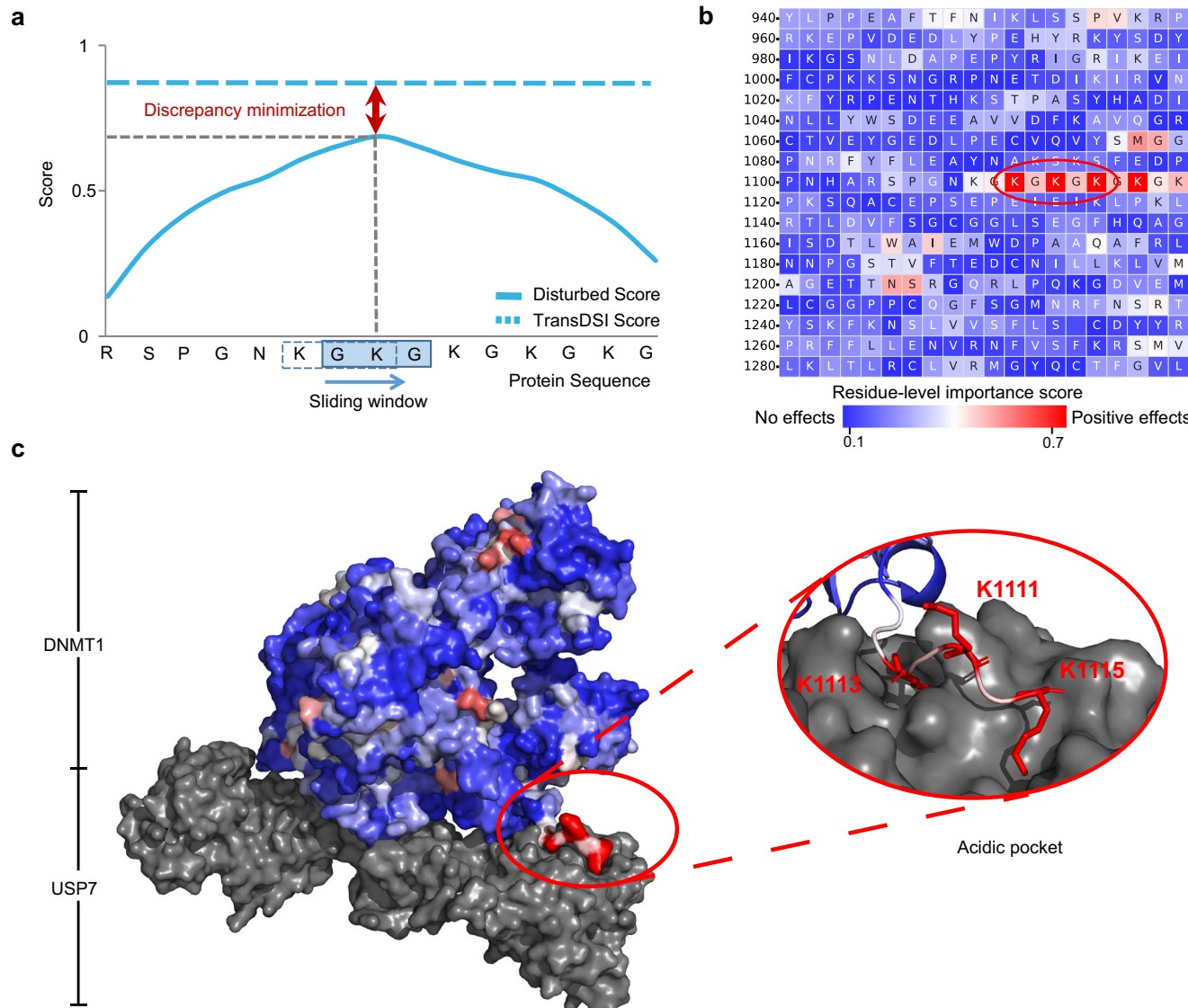

**Fig. 3 | Explainable module of TransDSI (PairExplainer) provides partial insights into the protein structural basis of the DSI of USP7-DNMT1.** **a** Schematic diagram of PairExplainer working principle: For both DUB and substrate, a sliding window of 3 residues moves along the protein sequence with a step size of 1 residue, performing in silico "knockout" by removing all residues except for the triad within the sliding window. The regions with the lowest change in "Disturbed Score" relative to "TransDSI Score" after knockout are more likely to be the binding sites for DSI (indicated by red arrows). **b** This heatmap visualizes the effect of each amino acid residue of DNMT1 (from 940 to 1,299) on USP7-DNMT1

interaction, with residues in red exerting positive effects and those in blue no effects. The Lysine residues (K1111, K1113, K1115) of DNMT1 (in the red oval box) harbor relatively higher scores in the importance map, which have been reported to be crucial for the interaction with USP7[27]. **c** Depiction of USP7-DNMT1 complex (PDB id: 4YOC) modeled in surface representation. Residue importance of the DNMT1 learned from PairExplainer with coloring ranging from low (blue) to high (red) are shown. Inset demonstrates details of USP7-DNMT1 interface, including the acidic pocket of USP7 and KG linker of DNMT1. Source data are provided as a Source Data file.

could serve as a potential therapeutic target for the treatment of tumors with FOXP3-induced immune evasion. USP20, on the other hand, has been shown to play important roles in various signaling pathways, such as enhancing the Wnt signaling pathway to promote tumor growth by deubiquitinating β-catenin[36]. However, no association between USP20 and autoimmune diseases has been reported so far. We found that USP20 can bind and deubiquitinate FOXP3 (Fig. 4d, e). This indicates that USP20 may play a role in regulating the autoimmune process and may be a potential target for inhibiting FOXP3-induced tumor immune escape.

In addition, we also predicted the DUBs of some upstream regulators of FOXP3 in TGF-β pathway. We predicted two DUBs that regulate SMAD3 (UCHL5, Score: 0.846) and SMAD4 (USP17, Score: 0.881). Literature review showed that these predictions were validated by independent studies[37,38] (Fig. 4a).

Candidate Substrates of USP22. Ubiquitin-specific protease 22 (USP22) has been implicated in the regulation of multiple signaling pathways (such as SIRT1/AKT/MRP1 signaling pathway) associated with the development and progression of HCC through its DUB activity[39]. Elevated USP22 expression is correlated with poor prognosis in HCC patients[40], positioning it as a potential therapeutic target for HCC intervention. The peptide of hD1 has been identified as a specific inhibitor of USP22[41]. However, the clinical applicability of USP22 as a drug target in patients remains to be elucidated. Clinical practice with certain targeted drugs, such as PD-1/PD-L1 inhibitors, has demonstrated that the efficacy of these agents is intricately linked to the cellular regulatory networks context of their targets[42]. Identifying the substrates of USP22 holds significant implications for understanding the pathogenesis of tumors and assessing the applicability of potential anticancer agents like hD1[41].

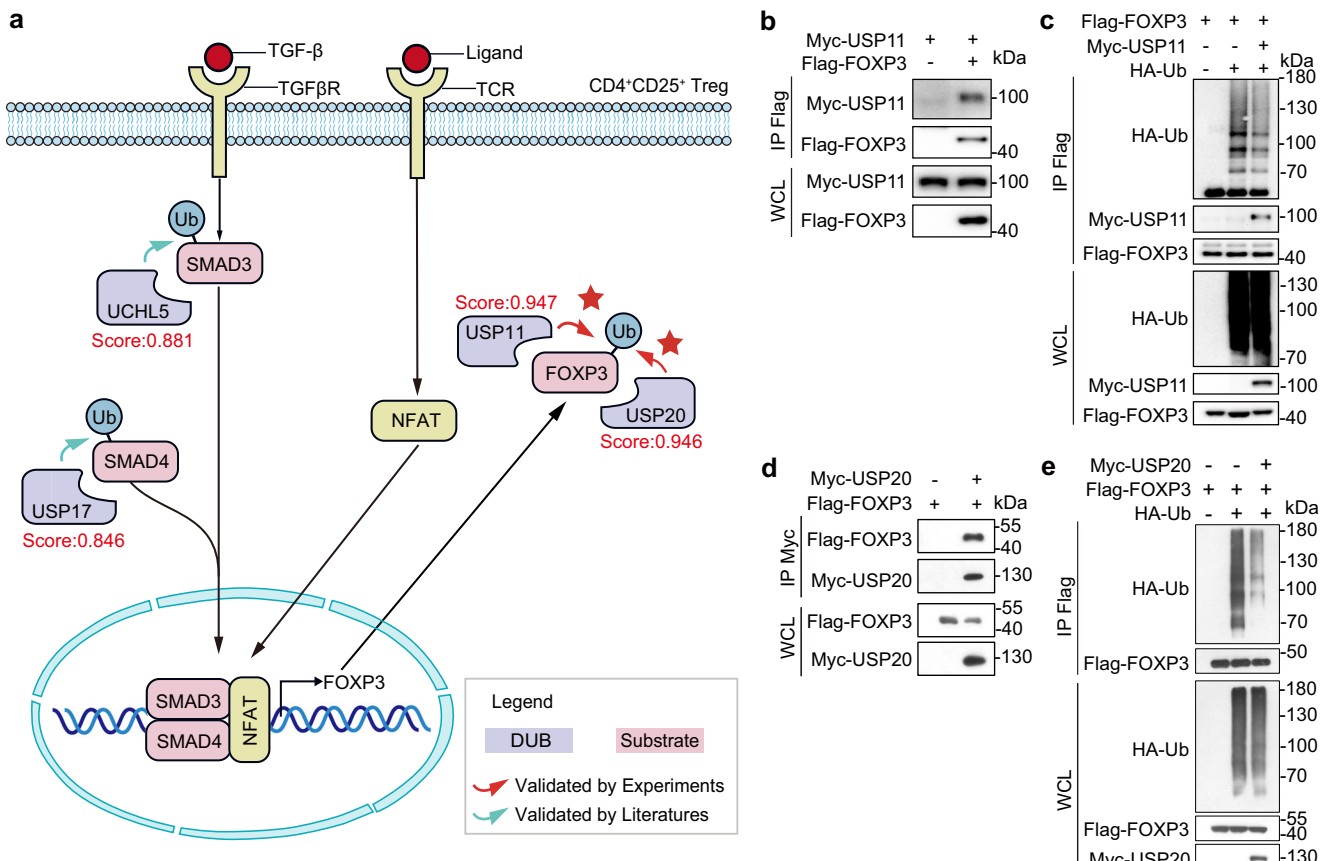

**Fig. 4 | Experimental validation of predicted DUBs (USP11 and USP20) for FOXP3. a** Prediction results of DUBs for both FOXP3 and upstream regulators (SMAD3/4) in the TGF-β pathway. SMAD3/4 and NFAT cooperate to induce *FOXP3* expression through binding to *FOXP3* enhancer[30]. Scores in the figure are assigned by TransDSI. The red pentagram indicates validation by our subsequent experiments. **b** Experimental validation of the USP11-FOXP3 interaction using Co-immunoprecipitation (IP) assay. HEK293T cells were transfected with Myc-USP11 and Flag-FOXP3, after 36 h, cells lysates were immunoprecipitated with indicated antibody and analyzed by western blot. **c** USP11 was found to remove ubiquitin modifications from FOXP3. HEK293T cells were transfected with HA-Ub, Myc-USP11, control vector or Flag-FOXP3, and cells were treated with MG132 for 8 h before collection. Ubiquitinated FOXP3 was immunoprecipitated with anti-Flag antibody and detected by immunoblotting with anti-HA antibody. **d** The USP20-FOXP3 interaction was validated by the same experimental validation protocol as the USP11-FOXP3 interaction in (**b**). **e** Like USP11 in (**c**) USP20 was also found to remove ubiquitin modifications from FOXP3, as determined by the same procedure. Source data are provided as a Source Data file.

Utilizing TransDSI, we have predicted 268 candidate substrates for USP22. We selected five DSIs with relatively high ranking for validation (CLSPN, score: 0.992; p53, score: 0.982; MDM4, score: 0.979; MDM2, score: 0.979; AR, score: 0.875). We observed that USP22 can act as a DUB to deubiquitinate both AR (Androgen receptor) and p53(Fig. 5a), both interactions were subsequently confirmed via exogenous co-immunoprecipitation (Co-IP) experiments, and USP22 decreased both AR (Fig. 5b, c) and p53 (Fig. 5f, g) ubiquitination in cells.

To further elucidate the clinical significance of the identified the deubiquitinating regulatory role of USP22 on AR, we classified 159 HCC patients with HBV infection into four distinct subgroups (G-I, G-II, G-III, and G-IV) based on USP22 and AR protein abundance in the tumor tissues (Fig. 5d; refer to Methods for further details). Notably, we found that subgroups exhibiting high USP22 expression did not present significant prognostic differences in comparison to those with low USP22 expression (G-I and G-II, log-rank $P = 0.53$; G-III and G-IV, log-rank $P = 0.92$; Fig. 5e). Intriguingly, a significant disparity in prognosis was observed within USP22 high-expression subgroups (G-II and G-III, log-rank $P$-value = 0.043, Fig. 5e). This demonstrates the significance of identifying USP22 substrates for achieving more precise subgrouping and promoting personalized treatment. Specifically, in G-II patients, USP22 might stabilize AR through deubiquitination, subsequently inhibiting HCC progression. Consequently, inhibition of USP22 may result in the downregulation of AR, attenuating its suppressive effect on HCC and rendering therapeutic strategies targeting USP22 unsuitable for this subgroup. And in G-III patients, AR stabilization was not mediated by USP22, suggesting that USP22-targeted therapy would not elicit AR-related side effects in this population.

In addition, the identified USP22-p53 interaction is of potential biological significance. p53 is a key tumor suppressor that inhibits excessive cell growth and division[43]. Our current understanding of the relationship between USP22 and p53 is limited to indirect associations. For example, Lin et al. (2012) proposed that USP22 can suppress *TP53* transcriptional activation by deubiquitinating SIRT1[39]. Our findings for the first time demonstrated that USP22 can directly deubiquitinate p53, providing a new perspective on the complexity of p53 regulation.

### Use cases of TransDSI in the analysis of disease omics data

Numerous types of high-volume omics datasets especially proteomic data (such as TCGA and CPTAC) are increasing exponentially with the advancement of high-throughput experimental techniques[44]. In-depth analysis of omics data often begins with exploring key molecules of interest identified from differential expression or survival analysis, to investigate their associations with disease prognosis[45,46]. In fact, the regulators of these key molecules are also very important for the study

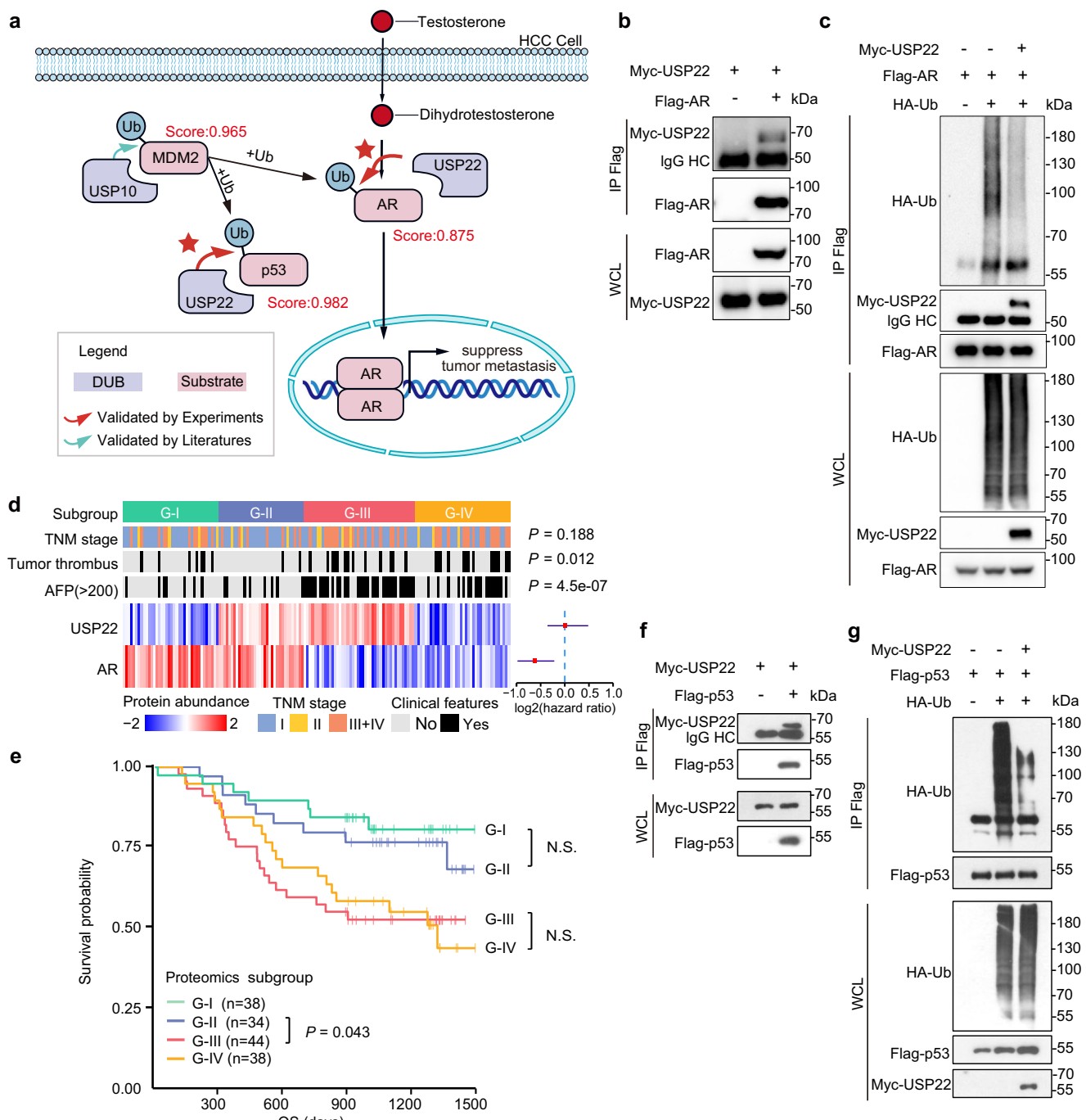

**Fig. 5 | Experimental validation of predicted substrates (AR and p53) of USP22.** **a** Prediction results of substrates of USP22. The homodimer of AR enters the nucleus to regulate the transcription of certain genes, such as *FKBP5*, inhibiting tumor metastasis. Scores in the figure are assigned by TransDSI. The red pentagram indicates validation by our subsequent experiments. **b** Experimental validation of the USP22-AR interaction with Co-IP assay. HEK293T cells were transfected with Myc-USP22 and Flag-AR, after 36 h, cells lysates were subjected to IP and western blot. **c** USP22 can stabilize AR through deubiquitination. HEK293T cells were transfected with HA-Ub, Myc-USP22, control vector or Flag-AR, and cells were treated with MG132 for 8 h before collection. Ubiquitinated AR was immunoprecipitated (IP) with anti-Flag antibody and detected by immunoblotting with anti-HA antibody. **d** HCC patient subgrouping ($n = 154$). Patients are grouped into four major subgroups (G-I, G-II, G-III and G-IV) based on the abundance of USP22 and AR, using a consensus-clustering analysis (Methods). Clinicopathologic factors including harboring tumor thrombus (*P*-value = 0.012) and higher AFP level (*P*-value = 4.5e-07) were more prominent in G-III and G-IV versus G-I and G-II (Kruskal-Wallis test). **e** Kaplan–Meier curves of overall survival (OS) for each protein subgroup (log-rank test). **f** Similarly, the USP22-p53 interaction was validated under the same validation protocol as the USP22-AR. **g** Like AR, p53 was also found to have its ubiquitin modifications removed by USP22, as determined by the same procedure. Source data are provided as a Source Data file.

of disease mechanisms and expanding the scope of omics analysis for discovering disease biomarkers and potential drug targets. TransDSI can facilitate the identification of potential regulators of these key molecules.

Specifically, we analyzed a cohort of 159 Chinese HCC patients with HBV infection (CHCC-HBV) from Fan's study[47]. We found that the M2 isoform of pyruvate kinase (PKM2), a key enzyme in the glycolytic pathway, is significantly upregulated in tumor compared to non-tumor

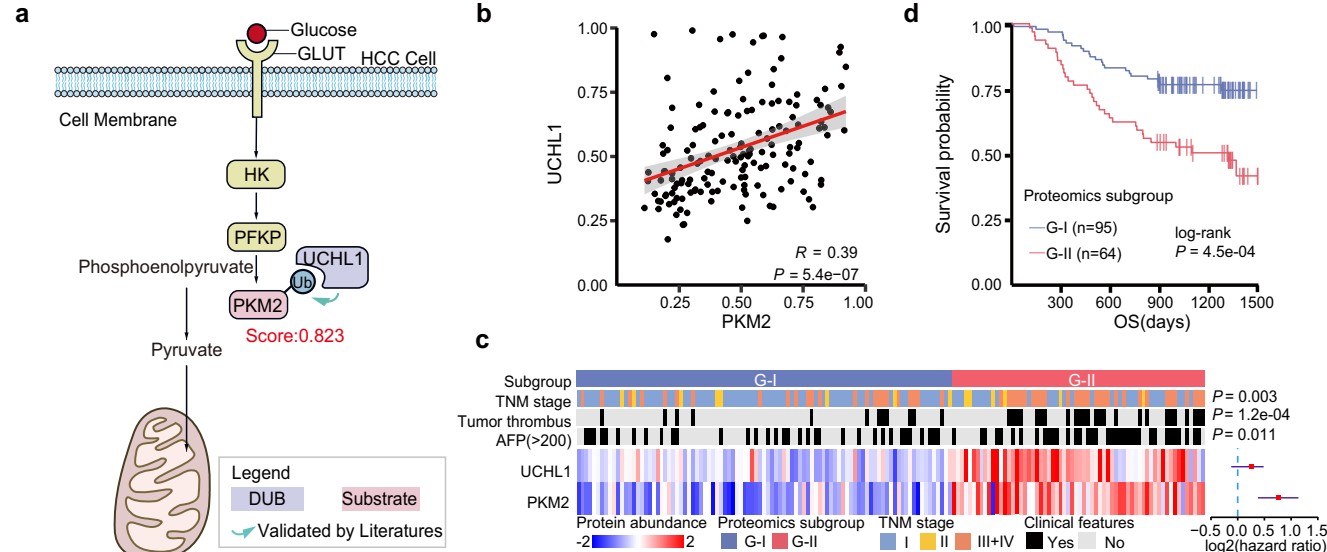

**Fig. 6 | Analysis of predicted DUBs for proteins involved in Glycolytic Pathway. a** PKM2 is a rate-limiting enzyme in the glycolytic pathway, which is highly upregulated in cancer cells and is associated with poor prognosis. UCHL1 was predicted to be regulator of this pathway, which was validated by independent studies[50]. The score shown in the figure is assigned by TransDSI. **b** Protein expression correlation between UCHL1 and PKM2 across 159 HCC patient samples (Pearson correlation coefficient = 0.39, P-value = 5.4e-07). P-value was calculated using a two-sided t-test. Shaded areas depict a 95% confidence interval. **c** HCC patient subgrouping. Patients are grouped into two major subgroups (G-I and G-II) based on the pairwise expression of UCHL1 and PKM2, using a consensus-clustering analysis (Methods). Clinicopathologic factors of these protein subgroups are evaluated, and patients in subgroup G-II were characterized by harboring tumor thrombus (P-value = 1.2e-04), higher AFP level (P-value = 0.011), and advanced TNM stages (P-value = 0.003) than those in G-I (Kruskal-Wallis test). Each column of the heatmap represents a patient sample and each row indicates a protein. The color of each cell represents the Z-score (log2 of the relative abundance, scaled by the protein's standard deviation) of the protein in that sample. **d** Kaplan–Meier curves of overall survival (OS) for each protein subgroup. The subgroup G-II showed a significantly lower overall rate of survival (P-value = 4.5e-04) and a greater risk of postoperative death than subgroup G-I (Log-rank test). Source data are provided as a Source Data file.

samples (adj. P-value = 1.3e-13, logFC = 0.78). Next, we aimed to associate PKM2 with other known tumor-related DUBs, so as to deeply understand the role of PKM2 deubiquitination regulatory in the mechanism of liver cancer[48]. Among the predicted DUBs for PKM2, ubiquitin carboxyl-terminal hydrolase L1 (UCHL1) emerged as a candidate due to its reported tumor-suppressive activity[49] and a high confidence score of 0.823 (Fig. 6a). In fact, recent independent studies have confirmed the above speculation in Parkinson's disease. Ham et al. found UCHL1 can stabilize PKM2 by mediating its deubiquitination, and loss of UCHL1 can reduce oxidative stress and alleviate neuronal damage by suppressing glycolysis[50]. However, the role of PKM2 deubiquitination in cancer development and progression remains elusive. Therefore, we next aim to explore the association between UCHL1-mediated PKM2 deubiquitination and tumor prognosis as well as overall survival.

Our analysis on CHCC-HBV revealed a significantly positive correlation between the expression of UCHL1 and PKM2 across all HCC samples (Fig. 6b, R = 0.39, P-value = 5.4e-07). Furthermore, through consensus-clustering analysis based on the abundance of UCHL1 and PKM2, we identified two major protein subgroups among the 159 HCC tumors, with 95 and 64 cases assigned to subgroups G-I and G-II, respectively (Supplementary Fig. 4a and c). The G-II subgroup is characterized by synergistic high expression of UCHL1 and PKM2. The enzymes upstream of PKM2 in the glycolytic pathway, HK and PFKP, are both highly expressed in the G-II type (Supplementary Fig. 4b). Clinical evaluation of these protein subgroups showed that patients in the G-II subgroup had a higher frequency of tumor thrombus (P-value = 1.2e-04), higher AFP level (P-value = 0.011), and advanced TNM stages (P-value = 0.003) compared to those in the G-I subgroup (Kruskal-Wallis test; Fig. 6c). Figure 6d shows that the G-II subgroup had a significantly lower overall survival rate compared to the G-I subgroup (log-rank test P-value = 4.5e-04). These findings suggest that overexpression of UCHL1 may stabilize PKM2 and result in metabolic

dysregulation and poor prognosis, and that inhibiting UCHL1 in the G-II subgroup may help improve patient outcomes by downregulating the over-expressed substrate PKM2.

In this study, we report a finding that contributes to understanding the mechanism through which UCHL1 may exert its oncogenic function in HCC. Our results suggest that UCHL1 may contribute to tumor progression in HCC by stabilizing PKM2 via deubiquitination, thereby providing important indications into the potential molecular mechanism of HCC pathogenesis. UCHL1, as the regulatory molecule for PKM2 proposed by TransDSI, may also be used as a potential drug target for resistant tumor treatment. This case study shows the usage of TransDSI for the disease omics dataset analysis, aiding the discovery of disease potential biomarkers or drug targets.

## Discussion

We established a protein sequence-based ab initio strategy, TransDSI, for predicting deubiquitinase-substrate interactions. This study transfers proteome-scale evolutionary information to predict potential DSIs. The performance of TransDSI outperforms multiple machine learning strategies against both cross-validation and independent test. Both bioinformatic analysis and experimental validation demonstrate the effectiveness of our strategy. In addition, as a general-purpose transfer learning model, the computational framework of TransDSI can alleviate the problem of insufficient training data and has the potential to predict other less-studied categories of protein-protein interactions, such as the case of SUMOylation.

Nevertheless, deep learning models constructed on relatively small training datasets remain prone to overfitting, this is because the model may only capture the noise in the training data, which is not present in the test data. As a result, the model may perform well on the training data, but poorly on an independent test set[51,52]. To avoid the possible overfitting, we have taken multiple strategies: (1) Protein embedding: We use a self-supervised learning module that is

completely independent of the DSI prediction task to learn protein feature representations[53]; (2) Neural network training: We use a variety of regularization techniques to prevent overfitting in the neural networks of the protein embedding module (Fig. 1b) and the DSI-predictor (Fig. 1c), including batch normalization and dropout[54]; (3) Model evaluation: To simulate the real-world use case, we use an independent test set to evaluate the performance of the DSI model. All DSIs contained in this independent test set were discovered after June 1, 2018, and none of the pairs of DSIs were used for training[55]; (4) Gold standard negative data set construction: The negative data set we construct has a similar network topology to the positive data set, which can effectively prevent overfitting[56,57]; (5) ESI prediction task: We adopted the same TransDSI deep learning framework to successfully establish a prediction system for protein ubiquitination ligase E3-substrate interaction(Supplementary Fig. 2 and 3), which demonstrates that the TransDSI deep learning framework has certain generalization ability.

Our previous UbiBrowser 2.0[14], relies on hand-crafting discriminant features or rules for proteins, its scalability is hindered by the bottleneck of feature engineering. UB2 cannot be implemented if the required features (homology, enriched protein domain and function, PPI network topology, and inferred DUB recognition consensus motif) about the proteins are not available, which greatly limits its application. In TransDSI, we only use protein sequence information, which is the most basic property of proteins, therefore TransDSI can be easily implemented at proteome level. Specifically, two protein sequence-based features (CT-encoded protein feature vectors and SSN based on sequence similarity) are integrated into TransDSI. Removing either the SSN or CT-encoded sequence information significantly reduced the model's predictive performance (Supplementary Fig. 5), which suggested both features synergistically enhance prediction performance by capturing distinct aspects: CT-encoded features might capture local sequence features of proteins, while SSN might capture evolutionary correlations between diverse proteins.

The PairExplainer module within TransDSI can identify protein sequence features that reflect associations between DUBs and substrates, and some of these features provide partial insights into the structural basis of DSI, contributing to the ubiquitin-proteasome system (UPS)-related drug design and cancer treatments. However, TransDSI was specifically designed for constructing a proteome-wide DSI network, and not all features from PairExplainer are the decisive factors for the binding, and some DSIs might involve a complex interplay between multiple protein regions. Additionally, due to the limited number of reported experimentally validated DSI binding site data (currently only 9 sites, involving 1 DUB and 5 substrates), predicting DSI binding sites remains a major challenge. In fact, we tested the possibility of incorporating rich ubiquitination site datasets from mass spectrometry into DSI prediction. However, our analysis did not reveal any association between ubiquitination sites and DSI binding sites, which suggests this information may not sufficiently aid in the accurately differentiation between DSIs and general PPIs (Supplementary Fig. 6). In the future, a deeper understanding of DSIs might enable the use of protein sequence features surrounding ubiquitination modification sites for DSI prediction.

In order to elucidate the value of TransDSI, we analyzed several prediction results from multiple perspectives, such as drug target discovery and personalized cancer treatment and disease omics data analysis. Firstly, in the aspect of drug target discovery and personalized treatment, Huang, X. et al. posited that protein levels in the human body are stringently regulated by the UPS, and dysregulation may lead to diseases such as cancer, making it a potential drug target for personalized cancer treatment[58]. Moreover, many oncogenes cannot be directly targeted, rendering the identification of DUBs that regulate them essential for cancer therapy. In this paper, we predicted

and experimentally validated two predicted DUBs (USP11 and USP20) for FOXP3, along with two predicted substrates (AR and p53) for USP22. Among them, USP22-AR provides new insights for precise subgrouping and individualized treatment strategies for HCC patients. The deubiquitination of p53 by USP22, as revealed in our study, offers a new perspective on the complexity of p53 regulation. Meanwhile, USP11/USP20-FOXP3 presents potential therapeutic target for inhibiting FOXP3-induced tumor immune evasion, as FOXP3 itself cannot be targeted due to its important role in autoimmunity[34]. Secondly, in the context of disease omics data analysis, Barabási, A. L. et al. advocated for a network-based approach to investigate human diseases, emphasizing the inter-regulation of biomolecules to reveal the complexity and diversity of diseases, as opposed to solely focusing on individual dysregulated proteins in omics data[59]. Inspired by this idea, we predicted that UCHL1 could function as a DUB for PKM2, which exhibits significant upregulation in the HCC protein expression profile. Patient survival information also suggested that there is a relationship between the UCHL1-PKM2 interaction and tumor development, providing new insights into the development of HCC.

To enhance the application of our strategy, we have compiled all the predicted DSIs as a dataset, which contains 19,461 DSIs between 85 DUBs and 5,151 potential substrates. These DSIs, along with some key regions on them and all the codes of TransDSI, are available on github (https://github.com/LiDlab/TransDSI). This resource can serve as a valuable tool for biologists to identify candidate DUBs and substrates for deubiquitination studies, and contribute to the understanding of the mechanism of UPS regulated protein homeostasis.

## Methods
### Gold standard positive dataset
A human DSI dataset, the Gold Standard Positive dataset (GSP), is sourced from our UbiBrowser 2.0 database[14] (Downloaded on March 1, 2022). This gold standard dataset was obtained by strict manual curation, and each pair of interaction is supported by traceable literature evidence. Its construction process is as follows: Firstly, we collected all the literature published from 1982 to 2021 that may involve DUB and substrate interactions from PubMed containing the following keyword combinations: ("deubiquitinase" OR "DUB") AND ("substrate" OR "substrates"). Then, we established a panel of three experienced experts to manually review these papers. Potential DUB-substrate interactions were manually filtered and verified based on the following textual patterns: "D deubiquitylates S...", "D mediates the deubiquitination of S...", "D targets S for deubiquitination...", "D stabilizes S...", "D suppresses the ubiquitination of S...", "D plays a crucial role in the deubiquitination of S...", "S is the substrate of D...", "S is deubiquitinated and stabilized by D...", where D is a DUB and S is a substrate. Finally, the GSP consists of 865 manually curated DSIs, each of which is annotated with supporting evidence for the deubiquitination relationship between the DUB and substrate (Supplementary Data 1). The dataset used for 5-fold cross-validation comprised 616 DSIs that had been validated by literature up to June 2018. The independent test set comprised 249 DSIs identified from June 2018 to August 2021.

### Gold standard negative datasets
We constructed the Gold Standard Negative dataset (GSN) based on the protocols of references Zahiri et al.[57]. and Yu et al.[60], with some modifications. Firstly, we obtained human physical PPI from BioGRID[61] (Released 25 January 2022). Next, we randomly select a negative set with the same number of nodes as that of the positive set from the complement graph of the known DSI network. The connectivity distribution of DUBs in the negative set network is the same as that in the positive (Supplementary Fig. 1). All interactions in negative set are the PPIs from BioGRID. None of the interactions in the GSN are present in the GSP.

## Protein sequence processing

In this study, protein sequences are derived from UniProt[62] (accessed on October 4, 2022) and encoded using the conjoint triad (CT) method[15], a widely applied method for encoding amino acid sequences in related fields. The CT method clusters 20 different amino acids into 7 classes based on the dipoles and volumes of the side chains, and the mapping between classes and amino acids is presented in Supplementary Data 8. The CT encoding captures the properties of a single amino acid and its adjacent amino acids by treating any three consecutive amino acids as a unit and counting their occurrence frequency within the protein sequence. This results in a fixed-dimension representation of the amino acid sequence with a dimensionality of 343 ($7 \times 7 \times 7$).

To construct a sequence similarity network (SSN), we used BLASTp (version 2.13.0+) to compare all reviewed human protein sequences in UniProt and screened for protein pairs with an E-value < 1e-4. The SSN is represented as an undirected graph, $SSN = (V,E,X)$, where $V$ represents the set of proteins in the SSN and $e_{ij} = <v_i,v_j> \in E$ represents an interaction in the SSN. The adjacency matrix SSM is used to represent the topological structure of the SSN, with the sequence identity score between the proteins serving as weights: $SSM_{ij}$ = sequence identity between $v_i$ and $v_j$ if $e_{ij} \in E$, otherwise $SSM_{ij} = 0$ (assuming that each node is not connected to itself and setting the diagonal elements to 0).

To preserve node-specific information and balance the contribution of node neighbors and the node itself in feature extraction, we normalized the SSM to obtain the normalized sequence similarity matrix $A$, which serves as the input to the GCN encoder.

$$\widetilde{A} = SSM + D_1 \tag{1}$$

$$A = D_2^{-1}\widetilde{A} + \widetilde{I} \tag{2}$$

$D_1$ and $D_2$ are the weighted degree matrix of SSM and $\widetilde{A}$ :

$$D_{1ii} = \sum_j SSM_{ij} \tag{3}$$

$$D_{2ii} = \sum_j \widetilde{A}_{ij} \tag{4}$$

$\widetilde{I}$ is a conditional identity matrix, where $\widetilde{I}_{ii} = \begin{cases} 1, if\, D_{2ii} = 0 \\ 0, if\, D_{2ii} \neq 0 \end{cases}$ (5)

## Protein sequence feature embedding (variational graph autoencoder model)

The TransDSI model is composed of three main components: a protein sequence feature embedding module (variational graph autoencoder model), a deubiquitinase-substrate interaction prediction module (DSI-Predictor), and an explainable module (PairExplainer).

VGAE is a machine learning technique for unsupervised feature extraction that generates latent representations that incorporate both network structure and node features[17]. The method trains an encoder and decoder in parallel.

The graph encoder is built using GCN and aims to project the protein sequence features $X$ onto the latent features $Z$, leveraging network evolutionary information represented in the graph-structured SSN matrix. The resulting numerical matrix of protein embeddings, $Z$, serves as an interpretable latent representation for undirected graphs learned using VGAE. This representation is used to effectively compress complex graph structure data in non-Euclidean space into simple, low-dimensional numerical vectors while preserving as much relevant information from the original input as possible.

We defined a spectral convolution function $f_{gcn}$[17]:

$$Z^{(l+1)} = f_{gcn}\left(Z^{(l)}, A|W^{(l)}\right) \tag{6}$$

Here, we utilized the input $Z^{(l)}$ in a convolutional operation, yielding the output $Z^{(l+1)}$. The normalized sequence similarity matrix $A$ serves as the kernel for this calculation. In our work, $Z^{(0)}$ is initialized as $X$. The individual layers of our graph convolutional network can be defined as follows:

$$f_{gcn}\left(Z^{(l)}, A|W^{(l)}\right) = \text{ReLU}(AZ^{(l)}W^{(l)}) \tag{7}$$

where $W^{(l)} \in \mathbb{R}^{d_l \times d_{l+1}}$. $d_l$ is the dimension of input for convolution, $d_{l+1}$ is the dimension of output after convolution. Our graph encoder consists of two GCN layers, and we let the prior over the latent variables $Z$ be the centered isotropic multivariate Gaussian[63]:

$$q(Z|X,A) = \prod_{i=1}^n q(z_i|X,A) \tag{8}$$

$$q(z_i|X,A) = \mathcal{N}\left(z_i|\mu_i, \text{diag}(\sigma_i^2)\right) \tag{9}$$

In our work, we defined the prior over the latent variables Z as a centered isotropic multivariate Gaussian distribution with mean $\mu$ and standard deviation $\sigma$.

$$z_i = \mu_i + \sigma_i \odot \epsilon_i \tag{10}$$

where $\odot$ is element-wise multiplication and $\epsilon_i \sim \mathcal{N}(0,1)$.

Next, we described a basic inner product decoder that aims to reconstruct $A$ using the learned latent variable $Z$:

$$p\left(\hat{A}|Z\right) = \prod_{i=1}^n \prod_{j=1}^n p(\hat{A}_{ij}|z_i,z_j) \tag{11}$$

$$p\left(\hat{A}_{ij} = 1|z_iz_j\right) = \text{sigmoid}\left(z_i z_j^T\right) \tag{12}$$

Finally, to maximize the similarity between the reconstructed sequence similarity $\hat{A}$ and the normalized sequence similarity matrix $A$, we optimized the model by minimizing the following loss function:

$$\mathcal{L} = \mathbb{E}_{q(Z|X,A)}\left[\log p(\hat{A}|Z)\right] - KL\left[q(Z|X,A) \parallel p(Z)\right] \tag{13}$$

In this study, the Kullback-Leibler (KL) divergence, $KL[q(\cdot) \parallel p(\cdot)]$, is employed to quantify the dissimilarity between the distributions $q(\cdot)$ and $p(\cdot)$[64]. As $p(Z)$ is assumed to follow a normal distribution with mean 0 and standard deviation 1 (i.e, $p(Z) \sim \mathcal{N}(0,1)$), the cost function represents the capability of the model in reconstructing the input network and aligning the latent variables with $p(Z)$. The optimization of the cost function with respect to the parameters of the encoder is performed using stochastic gradient descent.

## The deubiquitinase-substrate interaction prediction module (DSI-Predictor)

The DSI-Predictor module comprises of two components, a GCN encoder and an MLP. The GCN encoder is identical to the GCN encoder in the VGAE module. The MLP component is described in further detail below.

MLP is widely recognized as a powerful and prevalent method for supervised learning[65]. In our study, protein embeddings obtained from the DUBs and their substrates are concatenated and inputted into a 4-layer fully connected neural network. We defined the function $f_{mlp}$ as

follows:

$$P^{(l+1)} = f_{mlp}\left(P^{(l)} | W^{(l)}, b^{(l)}\right) \qquad (14)$$

In this study, we utilized an MLP where $P^{(l)}$ serves as the input and $P^{(l+1)}$ represents the output of each MLP layer. The matrix of filter parameters, $W^{(l)}$, and the bias of each layer, $b^{(l)}$, are learned during the training process. The operation performed at each layer of the MLP can be defined as follows:

$$f_{mlp}\left(P^{(l)} | W^{(l)}, P^{(l)}\right) = \text{Mish}(W^{(l)}P^{(l)} + b^{(l)}) \qquad (15)$$

MLP in this study consists of four layers, with batch normalization and dropout implemented between each layer. $W^{(l)} \in \mathbb{R}^{d_l \times d_{l+1}}$, $d_l$ is the dimension of input, $d_{l+1}$ is the dimension of output.

The probability $P^{(4)}$ obtained from the MLP is transformed into a "TransDSI Score" for confidence assessment through the following sigmoid function.

$$\text{TransDSI Score} = \frac{1}{1 + e^{-P^{(4)}/T}} \qquad (16)$$

Here, the temperature parameter T was set to 2 to fine-tune the smoothness of the score distribution, and enhance the final TransDSI score's discriminative power[66].

## The explainable module of TransDSI (PairExplainer)

Graph Neural Networks (GNNs) are neural network models that incorporate the dependencies within a graph through message passing between the nodes of the graph[18]. However, it poses a challenge in terms of explainability because these models are complicated by combining both graph structure and node information. Several GNN explainable methods have been proposed to address this issue, but they mainly focus on explaining predictions made by GNNs for node classification and graph classification tasks[67,68].

In this work, we propose PairExplainer, a general approach to provide an explanation of node features for link predictions made by any GNN-based model. The underlying principle of PairExplainer is that retaining important node features should result in a prediction that is similar to the original prediction made by the GNN-based model.

Let $G$ denote a graph with edges $E$ and nodes $V$, where each node is associated with a $d$-dimensional feature representation. Given node pair $(v_i, v_j)$ and GNN model $\Phi$. $N_i$ and $N_j$ denote the $h$-hop neighbors of nodes $v_i$ and $v_j$, respectively. Our aim is to extract the neighborhood features $X_{N_i} = \{x_k | v_k \in N_i\}$, $X_{N_j} = \{x_k | v_k \in N_j\}$ and the corresponding subgraphs $G_{N_i}$ and $G_{N_j}$, that are crucial for the prediction result of the link between $v_i$ and $v_j$. These features and subgraphs impact the feature representation of $v_i$ and $v_j$ during the GNN aggregation process.

$$y_{ij} = \Phi\left(X_{N_i} \cup X_{N_j}, G_{N_i} \cup G_{N_j}\right) \qquad (17)$$

In the task of predicting DSIs, the specific DUB and substrate are denoted as $v_i$ and $v_j$, respectively. The protein sequence features of these enzymes and substrates are encoded using the CT method and are represented by the feature vectors $x_i$ and $x_j$. $G$ corresponds to the sequence similarity network (SSN).

Our model DSI-Predictor, denoted as $\Phi$, has been fine-tuned on a set of gold standard datasets. The goal of PairExplainer is to learn two feature masks $M_{v_i}, M_{v_j} \in \mathbb{R}^{1 \times d}$, which intuitively assess the importance of each feature in the prediction process. If a feature is not important, then masking it should not significantly decrease the prediction probability. The prediction probabilities after masking the features are

presented as follows:

$$\hat{y}_{ij} = \Phi(X_{N_i} \odot \sigma(M_{N_i}) \cup X_{N_j} \odot \sigma(M_{N_j}), G_{N_i} \cup G_{N_j}) \qquad (18)$$

The element-wise multiplication is denoted as $\odot$ and the sigmoid function, which maps the mask to the interval [0, 1], is denoted as $\sigma$. The model is optimized using the loss function defined as follows:

$$\mathcal{L} = \left| y_{ij} - \hat{y}_{ij} \right| - \frac{\alpha}{2d} \sum_{k=1}^{2d} m_i - \frac{\beta}{2d} \sum_{k=1}^{2d} (m_k \cdot \log(m_k)$$
$$+ (1 - m_k) \cdot \log(1 - m_k)) \qquad (19)$$

Here $\alpha$ and $\beta$ are hyper-parameters that balance the loss function, which control the contribution of each term in the optimization process, $m_k \in M_{N_i} \cup M_{N_j}$.

## TransDSI model hyperparameters setting

The protein sequence feature embedding module employs a VGAE model, which consists of a two-layer GCN encoder[17]. The input and output dimensions of each GCN layer are set to 343. The optimization algorithm used is Adam with an initial learning rate of 1e-4, and a dropout rate of 0.1. The VGAE model is trained for 100 epochs.

The DSI-Predictor module consists of both a GCN encoder and an MLP. The parameters of the GCN encoder are transferred from the VGAE model, and the hyperparameters are set to the same values as those in the VGAE. The four layers of the MLP have input and output dimensions of 686, 512, 256, 64 and 1, respectively. The optimization algorithm is Adam with an initial learning rate of 1e-4, a dropout rate of 0.4, and a batch size of 64. The DSI-Predictor module is trained for 100 epochs.

In the PairExplainer module, the optimization algorithm is Adam with an initial learning rate of 0.01, and the model is trained for a total of 10,000 epochs. The hyper-parameters $\alpha$ and $\beta$ are set to 1 and 0.5 respectively for the loss calculation.

## Determining the optimal classification threshold

The Youden index[21] was used to determine the optimal classification threshold for TransDSI score (0.701) by balancing the sensitivity and specificity of the model (Youden index = sensitivity + specificity − 1). We calculated the Youden index against different thresholds, and we chose the threshold that maximizes the Youden index as the recommended cutoff. DSIs with scores > 0.701 are considered to be of high reliability. TransDSI score is a score between 0 and 1, with higher scores indicating higher reliability. Meanwhile, we also provide those predicted DSIs with scores smaller than the recommended cutoff in the Supplementary Data 4. Users can fine-tune the threshold based on their needs: lowering it for increased sensitivity (more clues) or raising it for enhanced specificity (more accurate results).

## Five-fold cross-validation

To test the efficacy of the overall performance of various assessment models, the 5-fold cross-validation protocol was used. The DSI dataset used for 5-fold cross-validation comprised 616 DSIs that had been validated by literature up to June 2018. This positive dataset and the corresponding negative datasets were randomly divided into five approximately equal subsets. The model was then trained and tested five times, with each fold serving as the test set once and the remaining four folds used for training. This process was done in turn five times, and finally the numbers of TPs and FPs against different thresholds across five test data sets were summed to calculate the TP/FP ratio, and the sensitivity (TP/T) and specificity (1-FP/F) for the ROC curve. Finally, we added the number of TP (true positive), FP (false positive), TN (true negative), and FN (false negative) obtained from the five tests against all possible thresholds to calculate the sensitivity (TP/T) and specificity

(1-FP/F) in the ROC curve, and the precision (TP/(TP + FP)) and recall (TP/(TP + FN)) in the PR curve. Then, we used the Youden Index method to find the optimal threshold for all methods[21] and based on this threshold, we calculated NPV (TN/(TN + FN)), PPV (TP/(TP + FP)), and F1-score to evaluate the predictive performance of the model. In each training session, the training set and the validation set are independent of each other, so the validation set will not be used for performance tuning.

## Bioinformatics analysis on DSIs

The Gene Ontology (GO)[23,24] enrichment analysis is conducted utilizing the R package clusterProfiler, encompassing DUBs, all known substrates, and all predicted substrates. Subsequently, an in-depth investigation is performed on the GO terms exhibiting significant enrichment for both DUB and its known or predicted substrates (Supplementary Data 5).

We use GO semantic similarity to measure the functional association between DUB and substrate. To quantify the semantic similarity between the GO terms of two proteins, we utilized the R package GOSemSim[69,70].

A protein-protein interaction (PPI) network is constructed using the interactions recorded by BioGRID (version: BIOGRID-ALL-3.4.159)[71]. To measure the neighborhood similarity between two proteins in the PPI network, we calculated the P-value from a Fisher's exact test on the overlap of their network neighbors:

$$PPI.NS = -\log_{10}P \qquad (20)$$

## HCC protein subgrouping analysis based on predicted DSIs

We obtained an HCC proteomics dataset ($n = 159$) from the Clinical Proteomic Tumor Analysis Consortium (CPTAC) of the National Cancer Institute[47]. The data is normalized using median standardization across all proteins to account for sample loading differences. Consensus clustering is performed using the R package ConsensusClusterPlus[47]. The protein abundance of predicted DSI (UCHL1-PKM2) is subjected to k-means consensus clustering respectively with the following parameters: 1,000 bootstraps repetitions, pItem = 0.8 (resampling 80% of any sample), pFeature = 1 (resampling all proteins), and k-means clustering with up to 6 clusters. Euclidean distance is used as the measure of sample clustering. The number of clusters is determined based on three criteria: the average pairwise consensus matrix within consensus clusters, the delta plot of the relative change in the area under the cumulative distribution function (CDF) curve, and the average silhouette distance for consensus clusters. The clustering results obtained using UCHL1-PKM2 indicated that k = 2 or k = 3 clusters are the optimal solutions for clustering, as evidenced by the average silhouette width, the consensus CDF, and delta plot. The k = 2 cluster consensus matrix is deemed to have the cleanest separation among clusters and the lowest proportion of ambiguous clustering (PAC)[72]. Therefore, the HCC proteomic data is clustered into 2 groups (Supplementary Fig. 4a). Similarly, for USP22-AR, the k = 4 cluster was chosen as the final clustering results using PAM consensus clustering with Canberra distance metric (Supplementary Fig. 4c). In the CHCC-HBV cohort, the expression of AR was not detected in the samples of five patients (T385, T387, T391, T393, T395), so we removed these patients' samples when performing subgrouping analysis based on USP22-AR protein abundance. Survival analysis of patient stratification in different subgroups is performed using the consensus clustering results. The Log-rank test is used to compare survival outcomes between the two subgroups generated by proteomics clustering, and Kaplan−Meier survival curves are plotted using the R package ggsurvplot. Results with P-values < 0.05 are considered statistically significant. Log2(hazard ratio) of each protein is calculated using Cox proportional hazards regression analysis. The association between clinical information and protein subgroups is examined using the Kruskal-Wallis test for categorical data versus continuous data[47]. Pearson correlation test is used to calculate the correlation coefficient and P-value.

## Experimental validation protocols

Cell lines: Human embryonic kidney HEK293T cells was purchased from ATCC. HEK293T cells was cultured in dulbeccos modification of Eagles medium (DMEM, Gibco) supplemented with fetal bovine serum (FBS, Gibco) and penicillin/streptomycin at 37° C in a humidified 5% CO_2 incubator.

Plasmids, antibodies and cell transfection: Human DUB library was purchased from OriGene Technologies. Full-length AR, TP53 and FOXP3 were cloned into pFlag-CMV-2 vectors (MLCC, L3435) as indicated. The antibodies we used in this study are as follows: anti-Myc (MBL, Cat# M047-3, RRID:AB_591112, 1:2000 for IB, 1:500 for IP), anti-Flag (MBL, Cat# M185-3, RRID:AB_10950447, 1:1000 for IB, 1:500 for IP), and anti-HA (MBL, Cat# M180-3, RRID:AB_10951811, 1:1000 for IB). As secondary antibodies, goat anti-mouse IgG (H + L) was used (Jackson, Cat# 115-035-003, RRID: AB_10015289, 1:4000 for IB) and detection was done by the SuperSignal™ West Pico PLUS chemiluminescent Detection Reagent (ThermoFisher, 34577). Cells were transfected with various plasmids using TuboFect reagent (ThermoFisher, R0534) according to the manufacturer's protocol.

Immunoprecipitation: Cells were lysed with TNTE 0.5% (50 mM tris-HCl (pH 7.5), 150 mM NaCl, 1 mM EDTA, and 0.5% Triton X-100) with protease inhibitor (MCE). Immunoprecipitation was performed using the indicated primary antibody for 3 h and incubated with protein A/G agarose beads (Santa Cruz) overnight at 4° C, which were then washed with TNTE 0.5% buffer three times. The lysates and immunoprecipitates were analyzed by western blotting with the indicated antibodies.

Ubiquitination assay: For in vivo ubiquitination assay, HEK293T cells was transfected with various plasmids and treated with the proteasome inhibitor MG132 (20 μM; Sigma) for 8-10 h before collection. Cells were lysed in RIPA buffer (50 mM Tris-HCl (pH 7.5), 1% NP-40, 1% sodium deoxycholate, 10% glycerinum, 150 mM NaCl, 5 mM EDTA, and 0.1% SDS) and then incubated with the indicated primary antibody for 3 h and protein A/G agarose beads (Santa Cruz) overnight at 4 °C. After washing three times, the lysates and immunoprecipitates were analyzed by western blotting with the indicated antibodies.

The uncropped scans of blots in Fig. 4 and Fig. 5 have been included in the source data file ("Fig. 4b-e", "Fig. 5b, c", "Fig. 5f, g" sheets). The areas that have been cropped are indicated by dashed lines in these uncropped scans.

## Construction of TransESI model for predicting ubiquitin E3 ligase-substrate interactions

We constructed a model (TransESI) for predicting ESIs following the same protocol of TransDSI framework. Firstly, we constructed a gold standard positive dataset of ESIs using a manual curation approach. We split the gold standard ESI dataset into a training set and an independent test set. The training set comprised 2,367 ESIs identified before June 2018, while the independent test dataset comprised 472 ESIs identified from June 2018 to August 2021. Next, we constructed a gold standard negative dataset of ESIs based on PPIs from BioGRID (Released 25 January 2022). We randomly selected a negative set of the same size as the positive set from the complement of the known ESI network. During this process, we ensured that the connectivity distribution of E3s in the negative set was consistent with that in the positive set. Then, we designed the same deep transfer learning framework for ESI (Supplementary Fig. 2), trained TransESI model on ESI

data using a similar approach to TransDSI and tested it on an independent test dataset.

## Statistics and reproducibility

All statistical analyses were performed using R (version 4.0.2 and 4.1.3). ROC curves were plotted and smoothed, and the area under the curve (AUROC) and its 95% confidence interval are simultaneously calculated. To determine if there are nonrandom associations between two categorical variables, statistical significance was considered at $P$-value < 0.05 using the one-tailed Wilcoxon test and Hypergeometric test. Kruskal-Wallis test was used to analyze the clinical data. All survival analyses among the proteomic subtypes, used Kaplan–Meier method; $p$-values were calculated using the Log-rank test. Hazard ratio (HR) was calculated from Cox proportional hazards regression analysis. $P$-value < 0.05 was considered as significantly different. Five machine learning models (including RF, XGBoost, SVMs, LR and KNN) were implemented in scikit-learn v1.0.1. The experiments in this study were independently repeated at least three times. Similar results were obtained.

## Reporting summary

Further information on research design is available in the Nature Portfolio Reporting Summary linked to this article.

## Data availability

All relevant data supporting the key findings of this study are available within the article and its Supplementary Information files. The DUB-substrate interaction (DSI), protein-protein interaction (PPI), protein sequence data, GO annotation data, and HCC proteomics data used in this studyare available in the Zenodo database under accession code https://zenodo.org/records/10468044. These datasets were sourced from publicly available databases as follows: the Human DSI dataset from UbiBrowser 2.0 [http://ubibrowser.ncpsb.org.cn/], the Human PPI dataset from BioGRID [https://thebiogrid.org/], and the protein sequence data and GO annotation data from UniProt [https://www.uniprot.org/]. The HCC proteomic data used in this study were sourced from the CPTAC research and are accessible in the PDC database at https://pdc.cancer.gov/pdc/study/PDCO00198. The PDB entry 4YOC, which is utilized for visualizing the USP7-DNMT1 complex, can be accessed from the Protein Data Bank at [https://doi.org/10.2210/pdb4yoc/pdb]. The data generated in this study is provided in the Supplementary Data files. Source data are provided with this paper.

## Code availability

The source code of TransDSI is packaged as π-TransDSI and available at Github https://github.com/LiDlab/TransDSI and Zenodo https://zenodo.org/records/10866136[73].

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

## Acknowledgements

This research was funded by the National key Research and Development Program of China (2022YFC3401500 to C.C.; 2023YFF1204600, 2020YFE0202200 and 2021YFA1301603 to Dong Li) and the National Natural Science Foundation of China (32271518 and 32088101 to Dong Li).

## Author contributions

D.L. (Dong Li) and C.C. directed and designed research; Y.L. (Yuan Liu) and D.L. (Dianke Li) designed, implemented and evaluated the algorithm; Y.L. (Yuan Liu), D.L. (Dianke Li), S.X., Y.Q., Y.L. (Yang Li) and X.L. performed bioinformatics analysis and HCC protein subgrouping analysis; X.Z., L.Z. and C.C. performed experimental validation of predicted DSIs; D.L. (Dong Li), C.C., Y.L. (Yuan Liu), D.L. (Dianke Li) and X.K. wrote the manuscript. All authors have read and agreed to the published version of the manuscript.

## Competing interests

The authors declare no competing interests.
