## [Peer Review File · Nature Communications]

A protein sequence-based deep transfer learning framework
for identifying human proteome-wide deubiquitinase-
substrate interactionsReviewer #1 (Remarks to the Author):

This paper introduced a deep learning-based computational pipeline to predict deubiquitinase (DUB) substrate interactions (DSIs) solely utilizing information from protein sequences. In addition, the authors proposed a sliding window-based approach to highlight the contribution of different positions in protein sequence by masking sequence residues respectively, to identify hotspots that are potential DSI binding sites. More importantly, two pairs of DSIs (USP11-FOXP3 and USP22-AR) were experimentally validated. The presented results are convincing and the manuscript is well written. Nevertheless, there are several major issues that need to be clarified before considering the acceptance of this manuscript.

Major issues

1. Regarding the case study identifying DUB for FOXP3, there are four additional predicted DUBs for FOXP3 (Table S2), i.e., UCHL1 (score: 0.998), UCHL3 (0.998), USP18 (0.998), USP20 (0.997), with better or equal score, compared with the one USP11 (score: 0.997) demonstrated in the manuscript. What is the rationale to validate USP11-FOXP3 instead of the other four DUBs? If UCHL1/UCHL3/USP18/USP20-FOXP3 have not been published previously, it is worth experimental validation for a more complete evaluation.
2. Regarding the case study identifying substrates of USP22, AR (score: 0.980) only ranks 106 in all the predicted substrates of USP22 (Table S2). What is the rationale to select AR for experimental validation? CLSPN, TP53, MDM2 and MDM4 are four substrates with the full prediction score (1.000). Have they been reported? If not, it is worth validation too.
3. Section "Use cases of DeepDSI in the analysis of disease omics data" is not highly linked with the main topic of the manuscript, which could be moved to the last part of the results section. Additionally, UCHL1 ranks second (score: 0.955) as DUB targeting PKM2, while OTUB1 ranks as the top DUB (score: 0.981, Table S2), which also needs a justification.
4. Is there any recommendation for the cutoff of DeepDSI score, to indicate a high-confidence DSI? This is important for the end users, as aforementioned.
5. It would be exciting if mutants of predicted hotspot residues could be constructed for at least one of four exemplified genes (USP11-FOXP3 and USP22-AR), for validating the prediction of contributing residues to DSI by PairExplainer.
6. The performance of this computational pipeline was only compared with the prediction tool (UbiBrowser 2.0) developed by the same group, with a lack of comparison with the tools from other groups addressing the similar task.
7. Only one metric (AUROC) was used to evaluate model performance, additional metrics (e.g., AUPRC) may help a more complete evaluation.

Minor issues

1. USP11-FOXP3 score was labelled as 0.996 in the manuscript (p.10, line: 27) and Fig.5 A, while it is indicated as 0.997 in Table S2.
2. Several typos need to be corrected: "concentrated" -> "concatenated"; "For vivo ubiquitylation assay" -> "For in vivo ubiquitylation assay".

Reviewer #2 (Remarks to the Author):

The authors of this paper have undoubtedly made significant strides in the development of DeepDSI, a novel protein interaction prediction model with promising implications for the analysis of disease omics data. Their work represents a noteworthy contribution to the field, shedding light on the critical role of protein ubiquitylation and the potential applications of DeepDSI in understanding complex disease mechanisms. However, it is crucial to scrutinize the methodology and evaluation metrics to ensure the robustness and applicability of this model in real-world scenarios.

Revised Comments:

Positive Gold Standard:

The authors should be commended for their efforts in addressing the issue of false positive interactions reported in databases and studies. To further enhance the model's reliability, it is

recommended that the authors devise a robust strategy to filter out these false positives. This may involve incorporating additional filtering steps or considering the confidence levels of interactions in the gold standard.

Gold Standard Negative Datasets:

While the use of random sampling for selecting negative samples is a common practice, it is vital to acknowledge the potential bias associated with this method. In light of recent studies highlighting the limitations of random pairing, the authors should explore alternative approaches for constructing gold standard negative datasets (<https://doi.org/10.1016/j.ygeno.2013.05.006>; <https://doi.org/10.1093/bioinformatics/btq483>).

Imbalance Ratio: The authors are encouraged to carefully consider the issue of imbalance ratio in their analysis. Protein interaction prediction problems inherently suffer from imbalanced data, and assessing the model's performance across a range of imbalance ratios (negative/positive ratio) is crucial for a comprehensive evaluation. This would provide a more holistic view of the model's effectiveness and its ability to handle real-world scenarios.

Feature Importance:

While the authors have discussed the model's expandability and introduced a module for this purpose, there is a notable absence of feature importance analysis and discussion in the paper. Incorporating a feature importance assessment can offer valuable insights into the contributions of individual features and guide further model refinement and interpretation.

Performance Assessment:

Although the authors have employed ROC/AUC as the main performance metric, it is advisable to include other relevant metrics, such as precision, recall (sensitivity), F-measure, and, notably, positive predictive value (PPV) and negative predictive value (NPV). These metrics are of utmost importance when considering the practical application of the model in real-world biological or clinical settings, where the balance between true positives and false positives/negatives is crucial (doi: 10.3389/fpubh.2017.00307).

Cross Validation:

The paper lacks sufficient details regarding the 5-fold cross-validation process. It is essential to clarify whether validation data were used in each fold for performance tuning. Additionally, considering the use of an independent test set, the authors should provide insights into the validation strategy employed during model development to ensure the robustness of their results. Having an independent validation set beyond the test set would strengthen the credibility of the findings.

Reviewer #3 (Remarks to the Author):

The authors introduced a comprehensive deep learning framework for predicting DUB-substrate interactions (DSIs) and identifying potential DSI binding sites. To facilitate the development of the DSI-Predictor model, they established a Gold Standard Positive (GSP) dataset comprising 863 manually curated DSIs sourced from the UbiBrowser 2.0 database.

In the proposed DeepDSI framework, the Graph Convolutional Network (GCN) encoder of the DSI-Predictor was initialized using parameters transferred from the self-supervised module, generating protein embeddings. The pre-training of the DeepDSI model involved a sequence similarity network spanning 20,398 proteins, followed by fine-tuning using the GSP dataset's 863 experimentally validated DSIs. The prediction process utilized an MLP to derive the "DeepDSI Score" for each potential DSI candidate from the concentrated embeddings. Furthermore, DeepDSI integrated an explainable module to facilitate the identification of binding sites between DUBs and substrates. The overall design of DeepDSI is coherent and lucidly presented. Despite the authors' experimental validation of two DSI pairs, aimed at showcasing DeepDSI's effectiveness, there are notable issues that require attention in its current iteration.

Q1. During the deep learning model training phase, the authors exclusively considered 863 DSIs as the positive dataset and balanced this with an equal number of randomly selected PPIs as the negative dataset. This balanced approach was intended to ensure dataset equilibrium. However, the use of such a limited training dataset raises concerns about potential overfitting of the proposed deep learning model. Surprisingly, the manuscript lacks any discussion on how this issue was addressed.

Another critical concern pertains to the selection process of the negative dataset. Employing randomly selected negative samples can significantly impact model training. To address the potential bias introduced by this approach, it is recommended that the authors undertake a minimum of 30 iterations of negative dataset random selection. By doing so, the robustness and efficacy of the proposed model in accurately distinguishing between DSIs and general PPIs can be comprehensively evaluated.

Q2. While the proposed model showcases a well-structured design, it remains essential to evaluate its predictive performance in comparison to traditional machine learning (ML) methods. To establish the efficacy of DeepDSI in predicting DUB-substrate interactions, it is crucial to directly compare its performance against these established ML methods using the same dataset and features.

If DeepDSI demonstrates superior performance over these conventional ML methods in such a head-to-head comparison, it would serve as a promising testament to the model's effectiveness in the realm of DUB-substrate interaction prediction. This comparative analysis would provide a clear demonstration of DeepDSI's potential and its capacity to outperform existing techniques.

Q3. The authors' assertion in the Abstract that "the performance of DeepDSI exceeds state-of-the-art methods" could potentially lead to an exaggerated perception of DeepDSI's predictive prowess. It's important to note that the comparison in this work was limited to UbiBrowser 2.0, a database primarily designed for E3 ligase/DUB-substrate interactions in eukaryotic species. The DSI prediction was executed using a protocol similar to UbiBrowser 1.0, incorporating elements like Gene Ontology (GO) terms, domain pairs, DUB recognition consensus motifs, and PPI networks into a Bayesian model for predicting human DUB-substrate interactions.

Given the scarcity of experimentally validated DSIs, it's understandable why the authors opted to compare their model solely with UbiBrowser. However, to enhance the rigor of their evaluation, it is advisable for the authors to explore a broader range of machine learning methods for comparison. This should be done using an independent testing dataset. It's crucial that the authors explicitly clarify that the performance comparison was exclusively conducted with UbiBrowser 2.0 to provide accurate context to readers.

Q4. While the experimental validation of two pairs of predicted DSIs serves to showcase DeepDSI's effectiveness, the detection of DSI binding sites within this work presents challenges. Without annotated experimentally validated DSI binding site data, evaluating the accuracy of predicted binding sites between DUBs and substrates becomes complex. Even with the inclusion of MD simulation results for the USP7-DNMT1 interaction, the lack of annotated information for experimentally confirmed DSI binding sites impedes a comprehensive assessment of the predicted sites.

A point of confusion arises from Fig. 3, where the USP7-DNMT1 interaction is depicted without the ubiquitin component. It's crucial to include ubiquitin, which attaches to specific lysine residues and plays a pivotal role in illustrating how DUBs (like USP7) recognize ubiquitinated substrates (such as DNMT1). Regarding the significance of the 'KGK' motif as a potential binding site in the USP7-DNMT1 interaction, clarifying the location of ubiquitin (Ub) within the USP7-DNMT1 complex is important. Understanding where ubiquitination occurs can significantly inform the detection of DSIs' binding sites.

To enhance the precision of binding site detection for DSIs, it's recommended that the authors take into consideration the information about ubiquitination sites. This inclusion would contribute to a more accurate and comprehensive analysis of DSI binding interactions.

Q5. The data statistics within UbiBrowser 2.0 indicate a substantial disparity between the numbers of E3-substrate interactions (ESIs) (4,068) and DUB-substrate interactions (DSIs) (967). In light of this, it's highly advisable for the authors to apply a similar framework to the prediction of ESIs using the proposed DeepDSI model. The abundant and balanced dataset of ESIs can provide a robust foundation for developing and evaluating a deep learning-based model.

By adopting the same framework for ESIs, the authors would have the opportunity to comprehensively assess the effectiveness and capabilities of the proposed DeepDSI model. Since it performed well on DSI prediction, its application to ESIs should indeed yield promising results. The considerable dataset size for ESIs ensures a solid training foundation for the deep learning model, potentially bolstering its predictive performance.

It's important to emphasize that having substantial datasets for both model training and performance testing is paramount. This approach will provide more conclusive evidence regarding the efficacy of the proposed DeepDSI model. Expanding the model's evaluation to ESIs can enhance its overall credibility and applicability.

Response to Reviewer 1:

Comments for the Author:

This paper introduced a deep learning-based computational pipeline to predict deubiquitinase (DUB) substrate interactions (DSIs) solely utilizing information from protein sequences. In addition, the authors proposed a sliding window-based approach to highlight the contribution of different positions in protein sequence by masking sequence residues respectively, to identify hotspots that are potential DSI binding sites. More importantly, two pairs of DSIs (USP11-FOXP3 and USP22-AR) were experimentally validated. The presented results are convincing and the manuscript is well written. Nevertheless, there are several major issues that need to be clarified before considering the acceptance of this manuscript.

A: Thanks for your great comments.

The major significance of our deep learning framework is transferring proteome-scale evolutionary information to predict unknown DSIs based on insufficient training dataset. To highlight this point, we changed the name of our deep learning framework from DeepDSI to TransDSI.

In this revision, thanks to your and other two reviewers' great comments, we have made substantial revisions to our manuscript: 1) Two new added experimental validation results (USP20-FOXP3 and USP22-TP53); 2) More comprehensive bioinformatics evaluation for TransDSI framework; 3) Evaluation of generalization ability of TransDSI on an independent task (E3 ligase-substrate interaction prediction); 4) Evaluation of the explainable framework of TransDSI (PairExplainer) using all known experimentally confirmed DSI binding sites dataset. We believe these added results make the TransDSI framework more convincing.

As a new proteome-wide DSI prediction framework, we hope that TransDSI will attract wide attention, just like our UbiBrowser for E3 ligase-substrate prediction (*Nat Commun.* 2017, 8:347. Google scholar citation: 147), which has become a leading bioinformatics tool in the field of ubiquitination modifications (*Nat Rev Methods Primers.* 2021, 1:53), contributing to new findings in various aspects of protein function, such as disease occurrence (*Circulation.* 2020, 142:1190), cell development (*Cell Stem Cell.* 2020, 26:755), tumor immunity (*Cancer Res.* 2022, 82:114), and tumor metastasis (*Cancer Commun.* 2021, 41:1007).

Major issues:

Q1: Regarding the case study identifying DUB for FOXP3, there are four additional predicted DUBs for FOXP3 (Table S2), i.e., UCHL1 (score: 0.998), UCHL3 (0.998), USP18 (0.998), USP20 (0.997), with better or equal score, compared with the one USP11 (score: 0.997) demonstrated in the manuscript. What is the rationale to validate USP11-FOXP3 instead of the other four DUBs? If UCHL1/UCHL3/USP18/USP20-FOXP3 have not been published previously, it is worth experimental validation for a more complete evaluation.

A: Thanks for your great comments.

1. The rationale to study USP11-FOXP3 in previous manuscript

We tend to select DSIs with certain biological significance and high ranking for validation. Both FOXP3 and USP11 play critical roles in cancer development: FOXP3 can suppress anti-tumor immune responses by regulating Treg cells (*Cancer Lett.* 2020, 490:174), while USP11 can stabilize FOXP3 expression, thereby maintaining Treg function (*J Immunol.* 2019, 203:2388). However, the regulatory mechanism of USP11 on Treg cells is still unclear. Furthermore, "USP11-FOXP3" has a high TransDSI score (USP11 ranks fourth among all potential DUBs of FOXP3, **Supplementary Table 4**).

2. Validation of other potential DUBs for FOXP3

We experimentally tested all these four pairs of interactions (UCHL1/UCHL3/USP18/USP20-FOXP3) during this revision. We identified the interactions between USP20 and FOXP3, and that between UCHL3 and FOXP3 (**Figure R1-1**). And we further observed that USP20 can act as a DUB to deubiquitinate FOXP3 (**Figure R1-2**). We did not observe the deubiquitination of FOXP3 by UCHL1, UCHL3, or USP18 in our experiments. This is possibly because some deubiquitinations require specific biological contexts, and our experimental validation system may not be able to recapitulate all possible biological contexts (*Adv Sci.* 2021, 8:2002484).

Our newly identified regulation of USP20 on the autoimmune-related protein FOXP3 has potential biological significance. USP20 has been shown to play important roles in various signaling pathways, such as enhancing the Wnt signaling pathway to promote tumor growth by deubiquitinating β -catenin (*Cell Death Differ.* 2018, 25:1855). However, no association between USP20 and autoimmune diseases has been reported so far. Our finding that USP20 can bind and deubiquitinate FOXP3 suggests that USP20 may play a role in regulating the autoimmune process, and may be a potential target for inhibiting FOXP3-induced tumor immune escape.

We have added the experimental results of USP20-FOXP3 in the revised manuscript, and added the above speculation. Please refer to section "**Results** - Experimental validation of predicted DSIs and their application in oncology research", **Figure R1-1** and **R1-2** in this letter or **Fig. 4** in revised manuscript for details.

Figure R1-1. Experimental validation of the UCHL3/USP20-FOXP3 interaction.

(a) Experimental validation of the UCHL3-FOXP3 interaction. HEK293T cells were transfected with Myc-UCHL3 and Flag-FOXP3. After 36h, cells lysates were analyzed by western blot. (b) The USP20-FOXP3 interaction was validated under the same experimental conditions as (a). (c) USP20 was found to remove ubiquitin modifications from FOXP3. HEK293T cells were transfected with HA-Ub, Myc-USP20, control vector or Flag-FOXP3, and cells were treated with MG132 for 8h before collection. Ubiquitinated FOXP3 was immunoprecipitated (IP) with anti-Flag antibody and detected by immunoblotting with anti-HA antibody.

Figure R1-2. Experimental validation of predicted DUBs (USP11 and USP20) for FOXP3.

(a) Prediction results of DUBs for both FOXP3 and upstream regulators (SMAD3/4) in the TGF- β pathway. SMAD3/4 and NFAT cooperate to induce FOXP3 expression through binding to FOXP3 enhancer. Scores in the figure are assigned by TransDSI. (b) Experimental validation of the USP11-FOXP3 interaction using Co-immunoprecipitation (IP) assay. HEK293T cells were transfected with Myc-USP11 and Flag-FOXP3, after 36h, cells lysates were immunoprecipitated with indicated antibody and analyzed by western blot. (c) USP11 was found to remove ubiquitin modifications from FOXP3. HEK293T cells were transfected with HA-Ub, Myc-USP11, control vector or Flag-FOXP3, and cells were treated with MG132 for 8 h before collection. Ubiquitinated FOXP3 was immunoprecipitated with anti-Flag antibody and detected by immunoblotting with anti-HA antibody. (d) The USP20-FOXP3 interaction was validated by the same experimental validation protocol as the USP11-FOXP3 interaction in (b). (e) Like USP11 in (c), USP20 was also found to remove ubiquitin modifications from FOXP3, as determined by the same procedure. Source data are provided as a Source Data file. (This is **Fig. 4** in the revised manuscript).

Q2: Regarding the case study identifying substrates of USP22, AR (score: 0.980) only ranks 106 in all the predicted substrates of USP22 (Table S2). What is the rationale to select AR for experimental validation? CLSPN, TP53, MDM2 and MDM4 are four substrates with the full prediction score (1.000). Have they been reported? If not, it is worth validation too.

A: Thanks for your great comments.

1. The rationale to select AR for experimental validation

As mentioned in our response to Q1, we chose USP22-AR for validation because both USP22 and AR are critical molecules in the progression of HCC (*Mol Oncol.* 2017, 11:682; *Endocr Relat Cancer.* 2014, 21:R165). Additionally, USP22 ranks first among AR's potential DUBs (1/87), and AR ranks among the top candidates for USP22 substrates (0.53%, 106/20398, **Supplementary Table 4**).

2. Validation of other potential substrates for USP22

In this revision, we have experimentally tested these four potential substrates for USP22 (CLSPN/TP53/MDM2/MDM4). The results showed that all of CLSPN, TP53, and MDM2 can interact with USP22 (**Figure R1-3** in this letter). We further observed that USP22 could deubiquitinate TP53. (**Figure R1-4** in this letter)

We have added the experimental validation results for USP22-TP53, and have expanded the discussion above. Please refer to section "**Results** - Experimental validation of predicted DSIs and their application in oncology research", **Figure R1-3** and **Figure R1-4** in this letter or **Fig. 5** in revised manuscript for details.

Figure R1-3. Experimental validation of the USP22-TP53/MDM2/CLSPN interaction.

(a) Experimental validation of the USP22-TP53 interaction. HEK293T cells were transfected with Myc-USP22 and Flag-TP53, after 36h, cells lysates were analyzed by western blot. (b-c) Similarly, the USP22-MDM2/CLSPN interaction was validated under the same experimental conditions. (d) USP22 was found to remove ubiquitin modifications from TP53. HEK293T cells were transfected with HA-Ub, Myc-USP22, control vector or Flag-TP53, and cells were treated with MG132 for 8 h before collection. Ubiquitinated TP53 was immunoprecipitated (IP) with anti-Flag antibody and detected by immunoblotting with anti-HA antibody.

Figure R1-4. Experimental validation of predicted substrates (AR and TP53) of USP22.

(a) Prediction results of substrates of USP22. The homodimer of AR enters the nucleus to regulate the transcription of certain genes, such as FKBP5, inhibiting tumor metastasis. Scores in the figure are assigned by TransDSI. (b) Experimental validation of the USP22-AR interaction with Co-IP assay. HEK293T cells were transfected with Myc-USP22 and Flag-AR, after 36h, cells lysates were subjected to IP and western blot. (c) USP22 can stabilize AR through deubiquitination. HEK293T cells were transfected with HA-Ub, Myc-USP22, control vector or Flag-AR, and cells were treated with MG132 for 8 h before collection. Ubiquitinated AR was immunoprecipitated (IP) with anti-Flag antibody and detected by immunoblotting with anti-HA antibody. (d) HCC patient subgrouping. Patients are grouped into four major subgroups (G-I, G-II, G-III and G-IV) based on the abundance of USP22 and AR, using a consensus-clustering analysis (Methods). The clinical characteristics of these protein subgroups are evaluated and late-stage disease such as larger tumor thrombus (P -value = 0.012) and high AFP status

(P-value = 4.5e-07) were more prominent in G-III and G-IV versus G-I and G-II (Kruskal-Wallis test). (e) Kaplan-Meier curves of overall survival (OS) for each protein subgroup (log-rank test). (f) Similarly, the USP22-TP53 interaction was validated under the same validation protocol as the USP22-P53. (g) Like AR, TP53 was also found to have its ubiquitin modifications removed by USP22, as determined by the same procedure. Source data are provided as a Source Data file. (This is **Fig. 5** in revised manuscript)

3. The issue of full prediction score (1.000)

The full prediction score of 1.000 in Table S2 of the previous manuscript is due to rounding errors in Excel software. The actual prediction scores for CLSPN, TP53, MDM2, and MDM4 are 0.99994, 0.99966, 0.99955, and 0.99955, respectively. Our model cannot produce a full prediction score of 1. The primary purpose of our scoring system is not just to assign scores, but more importantly, to provide a ranking of all candidates, with higher scores indicating greater reliability. In this revision, we have adjusted the prediction scores following the widely used approach by Hinton *et al.* (arXiv:1503.02531, 2015. This paper has been cited for 17311 times) to improve these scores' discriminatory power. The adjusted prediction scores for CLSPN, TP53, MDM2, and MDM4 are 0.992, 0.982, 0.979, and 0.979, respectively.

Q3: Section "Use cases of DeepDSI in the analysis of disease omics data" is not highly linked with the main topic of the manuscript, which could be moved to the last part of the results section. Additionally, UCHL1 ranks second (score: 0.955) as DUB targeting PKM2, while OTUB1 ranks as the top DUB (score: 0.981, Table S2), which also needs a justification.

A: Thanks for your great comments.

1. The rationale to select UCHL1 for further analysis

Here we intend to interpret the biological significance of the clues from the liver cancer proteomics data. We aimed to associate PKM2, a differentially expressed protein between liver cancer and adjacent tissues, with other known tumor-related proteins, so as to deeply understand the role of PKM2 in the mechanism of liver cancer (please refer to *Mol Biosyst.* 2013, 9:167 for such types of data analysis strategy).

We selected UCHL1 since it has been reported to inhibit HCC by stabilizing TP53 (*Hepatology.* 2008, 48:508). OTUB1 is ranked first, but its mechanism in liver cancer development is still unclear.

The interaction between UCHL1 and PKM2 has been validated by an independent study by Ham *et al.* (*Sci Adv.* 2021, 7: eabg4574). This makes our interpretation of the biomedical significance of PKM2 more convincing.

In our revised manuscript, we added the above speculation. Please refer to section "**Results** - Use cases of TransDSI in the analysis of disease omics data" and **Fig .6** in revised manuscript for details.

2. Position of "Use cases of TransDSI in the analysis of disease omics data"

Thanks for your suggestion. We have moved it to the last part of the **Results** section.

Q4: Is there any recommendation for the cutoff of DeepDSI score, to indicate a high-confidence DSI? This is important for the end users, as aforementioned.

A: Thanks for your great suggestion.

The issue of recommending a cutoff is important for end users to use our algorithm. We used the Youden index (*Biom J.* 2008, 50:419) to determine the recommended cutoff for TransDSI score. DSIs with scores greater than the recommended cutoff are considered to be of high reliability. TransDSI score is a score between 0 and 1, with higher scores indicating higher reliability.

The recommended cutoff is obtained by balancing the sensitivity and specificity of the model (Youden index = sensitivity + specificity - 1). We calculated the Youden index against different thresholds, and finally, we chose the threshold that maximizes the Youden index as the recommended cutoff.

Meanwhile, we also provide those predicted DSIs with scores smaller than the recommended cutoff in the **Supplementary Table 4**. Users can fine-tune the threshold based on their needs: lowering it for increased sensitivity (more clues) or raising it for enhanced specificity (more accurate results).

In the revised version, we added the above description. Please refer to Section "**Methods** - Determining the Optimal Classification Threshold" and **Supplementary Table 4** for details.

Q5: It would be exciting if mutants of predicted hotspot residues could be constructed for at least one of four exemplified genes (USP11-FOXP3 and USP22-AR), for validating the prediction of contributing residues to DSI by PairExplainer.

A: Thanks for your great comments.

TransDSI was specifically designed for constructing a proteome-wide DSI network. The explainable module of TransDSI (PairExplainer) can quantify the contribution of particular amino acid residues to DSI. In the revision, we collected all the DSI binding sites reported in the literature to comprehensively assess PairExplainer. We found that the PairExplainer module can identify sequence features that suggest associations between DUBs and substrates. Notably, some of these features provide partial insights into

the structural basis of DSI. For example, PairExplainer successfully captures the KxxxKxK motif on DNMT1 and UHRF1, which is known to bind USP7 and is one of only two recognized DUB recognition motifs (see **Table R1-1** in this letter and **Supplementary Table 7** in the revised manuscript for all these DSI site information). In **Fig. 3b**, we illustrated the KxxxKxK motif within the structure of the USP7-DNMT1 complex as determined by X-ray crystallography experiments (*Nat Commun.* 2015, 6:7023). Furthermore, PairExplainer successfully identified the MATH structural domain, which mediates the interaction between USP7 and MDM2/MDM4/TP53.

Crucially, although PairExplainer identified regions indicative of associations, these regions may not always exclusively dictate binding. For example, following your suggestion, a truncated FOXP3 mutant was constructed by deleting four amino acids (V283, A284, A285, G286). This mutant retains the ability to bind USP11 in HEK293T cells, suggesting that this specific region is not essential for DSI, and that the FOXP3-USP11 interaction may involve a complex interplay between multiple regions.

Additionally, due to the limited number of experimentally validated DSI binding site data (currently only 9 sites, involving 1 DUB and 5 substrates), predicting DSI binding sites remains a major challenge. We have revised the manuscript's description of PairExplainer to clarify this challenge.

Please refer to the section "**Results** - Explainable model of TransDSI provides partial insights into the protein structural basis of DSI" and **Supplementary Table 7** for further details.

Table R1-1. All experimentally confirmed DSI binding sites reported in the literature*.

DUB				Substrate				Source
Gene Symbol (DUB)	Substrate binding domain	Position	Identified by TransDSI	Gene Symbol (SUB)	DUB recognition motif	Position	Identified by TransDSI	
USP7	MATH domain	68-195	Yes	MDM4	P/AxxS	8-12	No	MEDLINE:20713061
USP7	MATH domain	68-195	Yes	MDM4	P/AxxS	398-402	No	MEDLINE:20713061
USP7	MATH domain	68-195	Yes	TP53	P/AxxS	359-363	No	MEDLINE:16402859
USP7	MATH domain	68-195	Yes	TP53	P/AxxS	364-368	No	MEDLINE:16474402
USP7	MATH domain	68-195	Yes	MDM2	P/AxxS	226-230	No	MEDLINE:16402859
USP7	MATH domain	68-195	Yes	MDM2	P/AxxS	147-151	No	MEDLINE:16474402
USP7	MATH domain	68-195	Yes	MDM2	P/AxxS	397-401	No	MEDLINE:20713061
USP7	ICP0-binding domain	622-801	No	DNMT1	KxxxKxK	1111-1115	Yes	MEDLINE:25960197
USP7	ICP0-binding domain	622-801	No	UHRF1	KxxxKxK	646-650	Yes	MEDLINE:26046769

*This table is **Supplementary Table 7** in the revised manuscript.

Q6: The performance of this computational pipeline was only compared with the prediction tool (UbiBrowser 2.0) developed by the same group, with a lack of comparison with the tools from other groups addressing the similar task.

A: Thanks for your great comments.

Our team has been engaged in establishing proteome-wide ESI/DSI networks since 2017 (*Nature communications*. 2017, 8: 347). Our UbiBrowser 2.0 is the only publicly available bioinformatics tool that can predict proteome-wide DSI (*Nucleic Acids Res.* 2022, 50: D719). In our previous manuscript, we compared TransDSI to UbiBrowser 2.0.

Thanks to your great comments, we constructed five additional DSI prediction systems employing machine learning methods including random forest (RF), support vector machine (SVM), eXtreme gradient boosting (XGBoost), logistic regression (LR), and K-nearest neighbors (KNN) (see section "**Results** - TransDSI has the ability to predict true DSIs"). To demonstrate the performance of the TransDSI deep transfer learning framework, we used the same training dataset and protein sequence features as that of TransDSI to construct these five DSI prediction systems. When evaluated against the same independent test set, TransDSI outperformed other prediction systems, exhibiting the largest area under the ROC curve (TransDSI: AUROC=0.75; RF: AUROC=0.70; XGBoost: AUROC=0.66; SVM: AUROC=0.66; LR: AUROC=0.66, **Figure R1-5**).

We have added the above results to the manuscript, which make our manuscript more convincing. Please refer to section "**Results** - TransDSI has the ability to predict true DSIs", **Figure R1-5** in this letter or **Fig. 2** and **Supplementary Table 2** in the revised manuscript for further details.

Figure R1-5. Performance evaluation of TransDSI. (a-d) Performance of various models (TransDSI, UB2, and five machine learning models) for the prediction of DSIs, evaluated through 5-fold cross validation and independent test, respectively. Five machine learning models are random forest (RF), support vector machine (SVM), eXtreme gradient boosting (XGBoost), logistic regression (LR), and K-

nearest neighbors (KNN). The ROC curves of the assessment models demonstrate sensitivity and specificity (**a, b**) and the PR curves of the assessment models precision and recall (**c, d**) against a particular prediction score cutoff, with each point on the curves representing the respective values. The 95% confidence intervals (95% CIs) of the sensitivity (precision) at the given specificity (recall) points are computed. The reference line indicates a non-informative prediction with an AUROC of 0.5 (**a, b**) or a prediction with a constant F1 score across different thresholds (**c, d**). (**e**) Multidimensional association features for known DSIs, predicted DSIs, and randomly screened DRIs. The feature scores include GO term similarity in terms of biological process (GO BP), cellular component (GO CC) and PPI network-based neighborhood similarity (PPI.NS). The colored bars represent the average value of various association features, with the error bars marking 95% confidence intervals. Wilcoxon test is used to test the difference between DRIs and known or predicted DSIs. Our results show that for all kinds of features, the average score of DRIs is significantly lower than that of known and predicted DSIs (* P-value < 0.05, *** P-value < 0.001). DRI, DUB-random protein interaction. (This figure is **Fig. 2** in the revised manuscript)

Q7: Only one metric (AUROC) was used to evaluate model performance, additional metrics (e.g., AUPRC) may help a more complete evaluation.

A: Thanks for your great suggestion.

Our previous manuscript only provided AUROC, which is a robust metric for evaluating model discrimination between positive and negative examples. Recognizing the importance of analyzing imbalanced data, we expanded the evaluation metrics in the revised version to include AUPRC, which excels in identifying true positives especially in imbalanced data sets (*ICML*. 2006, 233:240).

Additionally, thanks to your comments (Reviewer 2 also raised this point, Q5), we've incorporated F1-score, PPV, and NPV for a more comprehensive assessment. These metrics including PPV, NPV, and F1-score were calculated using a single, optimized threshold, and we determined this threshold by the Youden Index method (*Biom J*. 2008, 50:419). Notably, TransDSI consistently outperforms other methods across all metrics.

Please see all the evaluation results of all metrics in section "**Results** - TransDSI has the ability to predict true DSIs", **Supplementary Table 2, Figure R1-5** in this letter, or **Fig. 2** in the revised manuscript.

Minor issues:

Q8: USP11-FOXP3 score was labelled as 0.996 in the manuscript (p.10, line: 27) and Fig.5 A, while it is indicated as 0.997 in Table S2.

A: Thanks for your reminder. The two numbers are actually the same value (0.9969). The "0.997" in Table S2 of the previous manuscript is the result of Excel software rounding the decimal, while the value in **Fig. 5a** in the previous manuscript is the result of directly truncating the original data. In our revised manuscript, we have unified these numbers to avoid confusion.

Q9: Several typos need to be corrected: "concentrated" -> "concatenated"; "For vivo ubiquitylation assay" -> "For in vivo ubiquitylation assay".

A: Sorry for these mistakes. During the revision, we carefully checked the manuscript, and corrected these typos.

Response to Reviewer 2:

Comments for the Author:

The authors of this paper have undoubtedly made significant strides in the development of DeepDSI, a novel protein interaction prediction model with promising implications for the analysis of disease omics data. Their work represents a noteworthy contribution to the field, shedding light on the critical role of protein ubiquitylation and the potential applications of DeepDSI in understanding complex disease mechanisms. However, it is crucial to scrutinize the methodology and evaluation metrics to ensure the robustness and applicability of this model in real-world scenarios.

A: Thanks for your comments.

The major significance of our deep learning framework is transferring proteome-scale evolutionary information to predict unknown DSIs based on insufficient training dataset. To highlight this point, we changed the name of our deep learning framework from DeepDSI to TransDSI.

In this revision, thanks to your and other two reviewers' great comments, we have made substantial revisions to our manuscript: 1) Two new added experimental validation results (USP20-FOXP3 and USP22-TP53); 2) More comprehensive bioinformatics evaluation for TransDSI framework; 3) Evaluation of generalization ability of TransDSI on an independent task (E3 ligase-substrate interaction prediction); 4) Evaluation of the explainable framework of TransDSI (PairExplainer) using all known experimentally confirmed DSI binding sites dataset. We believe these added results make the TransDSI framework more convincing.

As a new proteome-wide DSI prediction framework, we hope that TransDSI will attract wide attention, just like our UbiBrowser for E3 ligase-substrate prediction (*Nat Commun.* 2017, 8:347. Google scholar citation: 147), which has become a leading bioinformatics tool in the field of ubiquitination modifications (*Nat Rev Methods Primers.* 2021, 1:53), contributing to new findings in various aspects of protein function, such as disease occurrence (*Circulation.* 2020, 142:1190), cell development (*Cell Stem Cell.* 2020, 26:755), tumor immunity (*Cancer Res.* 2022, 82:114), and tumor metastasis (*Cancer Commun.* 2021, 41:1007).

Q1: Positive Gold Standard:

The authors should be commended for their efforts in addressing the issue of false positive interactions reported in databases and studies. To further enhance the model's reliability, it is recommended that the authors devise a robust strategy to filter out these false positives. This may involve incorporating additional filtering steps or considering the confidence levels of interactions in the gold standard.

A: Sorry for the unclear description for the construction of gold standard positive (GSP) dataset in the previous manuscript.

The gold standard dataset we used was obtained by our manual curation, and each pair of interaction is supported by traceable literature evidence. Its construction process is as follows:

First, we collected literature abstracts that may involve DUB and substrate interactions from PubMed containing the following keyword combinations: ('deubiquitinase' OR 'DUB') AND ('substrate' OR 'substrates').

Then, we established a panel of three experts to manually review these papers. Potential DUB-substrate interactions were manually filtered and verified based on the following textual patterns: "D deubiquitylates S...", "D mediates the deubiquitination of S...", "D targets S for deubiquitination...", "D stabilizes S...", "D suppresses the ubiquitination of S...", "D plays a crucial role in the deubiquitination of S...", "S is the substrate of D...", "S is deubiquitinated and stabilized by D...", where D is a DUB and S is a substrate.

The above process has been published in our previous paper (*Nucleic Acids Res.* 2022, 50:D719). We also added simple description of this process in the revised manuscript. Please refer to section "**Methods** - Gold standard positive dataset" for details. We provide this gold standard positive dataset in **Supplementary Table 1**.

Q2: Gold Standard Negative Datasets:

While the use of random sampling for selecting negative samples is a common practice, it is vital to acknowledge the potential bias associated with this method. In light of recent studies highlighting the limitations of random pairing, the authors should explore alternative approaches for constructing gold standard negative datasets (<https://doi.org/10.1016/j.ygeno.2013.05.006>; <https://doi.org/10.1093/bioinformatics/btq483>).

A: Sorry for the unclear description in the previous manuscript about the Gold Standard Negative Datasets. In fact, we adopted the similar strategy to those presented in the literatures you provided (*Genomics*. 2013, 102:237; *Bioinformatics*. 2010,26:2610) (Figure R2-1 in this Cover Letter). These two papers are very representative in the field of constructing gold standard negative datasets. Their idea is to control the node connectivity distribution of the network represented by the negative set to be consistent with that of the network represented by the positive set, so as to ensure that two networks have similar network topological structures.

In this revised version, we added relevant method descriptions together with two references you mentioned. For details, please see “Methods - Gold standard negative datasets”, and the Supplementary Fig. 1 in the revised manuscript (Figure R2-1 in this cover letter).

Figure R2-1. Construction of the gold standard negative dataset. We constructed the negative set by randomly sampling an equal number of nodes from the complement graph of the known DSI network, ensuring identical DUB connectivity distributions between the negative and positive sets. All interactions within the negative set were derived from PPI data in the BioGRID database (*Nucleic Acids Res.* 2006, 34: D535). This figure is **Supplementary Fig. 1** in the revised manuscript.

Q3: Imbalance Ratio: The authors are encouraged to carefully consider the issue of imbalance ratio in their analysis. Protein interaction prediction problems inherently suffer from imbalanced data, and assessing the model's performance across a range of imbalance ratios (negative/positive ratio) is crucial for a comprehensive evaluation. This would provide a more holistic view of the model's effectiveness and its ability to handle real-world scenarios.

A: Thanks for your reminder.

Yes, the issue of imbalance ratio is very important for protein interaction prediction.

Our revised manuscript investigates the performance of TransDSI in comparison with five machine learning methods across diverse negative/positive ratios (1:1, 2:1, 5:1, and 10:1). As shown in **Supplementary Table 3 (in the revised manuscript)**, TransDSI consistently outperforms other methods at all ratios, demonstrating its ability to handle real-world scenarios.

We've added the above analysis and results to the section "**Results** - TransDSI has the ability to predict true DSIs" of the revised manuscript.

Q4: Feature Importance:

While the authors have discussed the model's expandability and introduced a module for this purpose, there is a notable absence of feature importance analysis and discussion in the paper. Incorporating a feature importance assessment can offer valuable insights into the contributions of individual features and guide further model refinement and interpretation.

A: Thanks for your reminder.

Assessing feature importance is an important issue for machine learning model construction, which is essential for further interpretation, optimization, and extension of the model. Following your great suggestion, we have incorporated a comprehensive analysis of model feature importance in the revised manuscript.

To independently evaluate the impact of both features used by TransDSI (CT-encoded protein feature vectors and SSN based on sequence similarity), we constructed two additional models: 1) TransDSI w/o SSN model: Utilizes CT-encoded sequence features while excluding SSN information; 2) TransDSI w/o CT model: Employs SSN features while eliminating CT-encoded data.

Against 5-fold cross-validation and an independent test, we found that removing either the SSN or CT-encoded sequence information significantly reduced the model's predictive performance. TransDSI,

which integrates both features, demonstrated the highest prediction efficacy among the three models (**Figure R2-2 in this letter**). This suggests that the two features synergistically enhance the prediction performance by capturing distinct aspects: CT-encoded features might capture local sequence features of proteins, while SSN might capture evolutionary correlations between diverse proteins.

We have added the above results and discussion in the revised version, please refer to the section "**Discussion**" of the manuscript and **Figure R2-2** in this letter (which is **Supplementary Fig. 4** in the revised manuscript) for further details.

Figure R2-2. Feature importance analysis for TransDSI. To independently evaluate the impact of both features used by TransDSI (CT-encoded protein feature vectors and SSN based on sequence similarity), we constructed two additional models: 1) TransDSI w/o SSN model: Utilizes CT-encoded sequence features while excluding SSN network information; 2) TransDSI w/o CT model: Employs SSN features while eliminating CT-encoded data. Employing 5-fold cross-validation and independent test, we found that removing either the SSN or CT-encoded sequence information significantly reduced the model's predictive performance. TransDSI, which encompasses both features, demonstrated the highest prediction accuracy among the three models. **(a-d)** Performance of three models (TransDSI, TransDSI

w/o SSN, TransDSI w/o CT) for the prediction of DSIs, evaluated through 5-fold cross-validation and independent test, respectively. The ROC curves of the assessment models demonstrate sensitivity and specificity (**a, b**) and the PR curves of the assessment models precision and recall (**c, d**) against a particular prediction score cutoff, with each point on the curves representing the respective values. The 95% confidence intervals (95% CIs) of the sensitivity (precision) at the given specificity (recall) points are computed. The reference line indicates a non-informative prediction with an AUROC of 0.5 (**a, b**) or a prediction with a constant F1 score across different thresholds (**c, d**). CT, conjoint triad; SSN, sequence similarity network. This figure is **Supplementary Fig. 4** in the revised manuscript.

Q5: Performance Assessment:

Although the authors have employed ROC/AUC as the main performance metric, it is advisable to include other relevant metrics, such as precision, recall (sensitivity), F-measure, and, notably, positive predictive value (PPV) and negative predictive value (NPV). These metrics are of utmost importance when considering the practical application of the model in real-world biological or clinical settings, where the balance between true positives and false positives/negatives is crucial (doi: 10.3389/fpubh.2017.00307).

A: Thanks for your great suggestion.

Our previous manuscript provided AUROC, which is a robust metric for evaluating model discrimination between positive and negative examples. Recognizing the importance of analyzing imbalanced data, we expanded the evaluation metrics in the revised manuscript to include AUPRC, which excels in identifying true positives especially in imbalanced data sets (*ICML*. 2006, 233:240).

Furthermore, thanks to your comment (Reviewer 1 also raised this point, Q7), we've incorporated F1-score, PPV, and NPV for a more comprehensive assessment. It's crucial to note that these added metrics including PPV, NPV, and F1-score depend on a single, optimized threshold, and we determined the threshold by the Youden Index method. Notably, TransDSI consistently outperforms other methods across all metrics.

Please see all the evaluation results of all metrics in section "**Results** - TransDSI has the ability to predict true DSIs", **Supplementary Table 2** in the revised manuscript, **Figure R2-3** in this letter (which is **Fig. 2** in the revised manuscript).

Figure R2-3. Performance evaluation of TransDSI. (a-d) Performance of various models (TransDSI, UB2, and five machine learning models) for the prediction of DSIs, evaluated through 5-fold cross validation and independent test, respectively. Five machine learning models are random forest (RF), support vector machine (SVM), eXtreme gradient boosting (XGBoost), logistic regression (LR), and K-

nearest neighbors (KNN). The ROC curves of the assessment models demonstrate sensitivity and specificity (**a, b**) and the PR curves of the assessment models precision and recall (**c, d**) against a particular prediction score cutoff, with each point on the curves representing the respective values. The 95% confidence intervals (95% CIs) of the sensitivity (precision) at the given specificity (recall) points are computed. The reference line indicates a non-informative prediction with an AUROC of 0.5 (**a, b**) or a prediction with a constant F1 score across different thresholds (**c, d**). (**e**) Multidimensional association features for known DSIs, predicted DSIs, and randomly screened DRIs. The feature scores include GO term similarity in terms of biological process (GO BP), cellular component (GO CC) and PPI network-based neighborhood similarity (PPI.NS). The colored bars represent the average value of various association features, with the error bars marking 95% confidence intervals. Wilcoxon test is used to test the difference between DRIs and known or predicted DSIs. Our results show that for all kinds of features, the average score of DRIs is significantly lower than that of known and predicted DSIs (* P-value < 0.05, *** P-value < 0.001). DRI, DUB-random protein interaction. (This figure is **Fig. 2** in the revised manuscript).

Q6: Cross Validation:

The paper lacks sufficient details regarding the 5-fold cross-validation process. It is essential to clarify whether validation data were used in each fold for performance tuning. Additionally, considering the use of an independent test set, the authors should provide insights into the validation strategy employed during model development to ensure the robustness of their results. Having an independent validation set beyond the test set would strengthen the credibility of the findings.

A: Thanks for your comments.

1. Details of 5-fold cross-validation process

Sorry for the absence of the 5-fold cross-validation process details in the previous version.

In the revision, we have provided the following details:

First, we randomly divided GSP (only DSIs identified before June 1, 2018) and the corresponding GSN into five approximately equal subsets.

Then, in each iteration of the model training process, we used one of these five subsets as the validation set each time, and the remaining four subsets were combined as the training set. This process was repeated five times.

Finally, we added the number of TP (true positive), FP (false positive), TN (true negative), and FN (false negative) obtained from the five tests against all possible thresholds to calculate the sensitivity (TP/T) and specificity ($1-FP/F$) in the ROC curve, and the precision ($TP/(TP+FP)$) and recall ($TP/(TP+FN)$) in the PR curve.

In each training session, the training set and the validation set are independent of each other, therefore, the validation set will not be used for performance tuning.

2. Independent validation dataset to evaluate TransDSI model performance

In evaluating the performance of our model, as well as comparing its performance with other models, we used a completely independent test dataset. The DSIs in this dataset were all identified between June 1, 2018 and August 1, 2021, and did not appear in the training dataset used in the 5-fold cross-validation process. This construction of such an independent validation set beyond the training set would strengthen the credibility of the findings.

Thanks for your reminder. We have added the above details about 5-fold cross-validation and independent test in the manuscript. Please refer to the section "**Methods** - Five-fold cross-validation" for further details.

Response to Reviewer 3:

Comments for the Author:

The authors introduced a comprehensive deep learning framework for predicting DUB-substrate interactions (DSIs) and identifying potential DSI binding sites. To facilitate the development of the DSI-Predictor model, they established a Gold Standard Positive (GSP) dataset comprising 863 manually curated DSIs sourced from the UbiBrowser 2.0 database.

In the proposed DeepDSI framework, the Graph Convolutional Network (GCN) encoder of the DSI-Predictor was initialized using parameters transferred from the self-supervised module, generating protein embeddings. The pre-training of the DeepDSI model involved a sequence similarity network spanning 20,398 proteins, followed by fine-tuning using the GSP dataset's 863 experimentally validated DSIs. The prediction process utilized an MLP to derive the "DeepDSI Score" for each potential DSI candidate from the concentrated embeddings.

Furthermore, DeepDSI integrated an explainable module to facilitate the identification of binding sites between DUBs and substrates. The overall design of DeepDSI is coherent and lucidly presented. Despite the authors' experimental validation of two DSI pairs, aimed at showcasing DeepDSI's effectiveness, there are notable issues that require attention in its current iteration.

A: Thanks for your high evaluation.

The major significance of our deep learning framework is transferring proteome-scale evolutionary information to predict unknown DSIs based on insufficient training dataset. To highlight this point, we changed the name of our deep learning framework from DeepDSI to TransDSI.

In this revision, thanks to your and other two reviewers' great comments, we have made substantial revisions to our manuscript: 1) Two new added experimental validation results (USP20-FOXP3 and USP22-TP53); 2) More comprehensive bioinformatics evaluation for TransDSI framework; 3) Evaluation of generalization ability of TransDSI on an independent task (E3 ligase-substrate interaction prediction); 4) Evaluation of the explainable framework of TransDSI (PairExplainer) using all known experimentally confirmed DSI binding sites dataset. We believe these added results make the TransDSI framework more convincing.

As a new proteome-wide DSI prediction framework, we hope that TransDSI will attract wide attention, just like our UbiBrowser for E3 ligase-substrate prediction (*Nat Commun.* 2017, 8:347. Google scholar citation: 147), which has become a leading bioinformatics tool in the field of ubiquitination modifications

(*Nat Rev Methods Primers*. 2021, 1:53), contributing to new findings in various aspects of protein function, such as disease occurrence (*Circulation*. 2020, 142:1190), cell development (*Cell Stem Cell*. 2020, 26:755), tumor immunity (*Cancer Res*. 2022, 82:114), and tumor metastasis (*Cancer Commun*. 2021, 41:1007).

Q1: During the deep learning model training phase, the authors exclusively considered 863 DSIs as the positive dataset and balanced this with an equal number of randomly selected PPIs as the negative dataset. This balanced approach was intended to ensure dataset equilibrium. However, the use of such a limited training dataset raises concerns about potential overfitting of the proposed deep learning model. Surprisingly, the manuscript lacks any discussion on how this issue was addressed. Another critical concern pertains to the selection process of the negative dataset. Employing randomly selected negative samples can significantly impact model training. To address the potential bias introduced by this approach, it is recommended that the authors undertake a minimum of 30 iterations of negative dataset random selection. By doing so, the robustness and efficacy of the proposed model in accurately distinguishing between DSIs and general PPIs can be comprehensively evaluated.

A: Thanks for your great comments.

1. Overfitting issues

As you pointed out, overfitting is a critical problem in machine learning, especially when the training data is small. This is because the model may only capture the noise in the training data, which is not present in the test data. As a result, the model may perform well on the training data, but poorly on an independent test set (References: *Neural comput*. 1992, 4:1; *J Chem Inf Comput Sci*. 2004, 44:1).

Because the training set used in TransDSI is limited as well, we have taken the following strategies to avoid overfitting:

1) Protein embedding: We use a self-supervised learning module that is completely independent of the DSI prediction task to learn protein feature representations to reconstruct the sequence similarity network (SSN). (Reference: *Bioinformatics*. 2020, 36:1234)

2) Neural network training: We use a variety of regularization techniques to prevent overfitting in the neural networks of both the protein embedding (**Fig. 1b**) and the DSI-predictor modules (**Fig. 1c**), including batch normalization and dropout. (Reference: *ICML*. 2015, 448:456)

3) Model evaluation: To simulate the real-world use case, we use an independent test set to evaluate the performance of the DSI model, and compare TransDSI with other algorithms. All DSIs contained in this independent test set were identified after June 1, 2018, and none of the pairs of DSIs appear in the training set of the model. (Reference: *Nat Methods*. 2016, 13:703)

4) Gold negative data set construction: The negative data set we constructed has a similar network topology to the positive data set, which can effectively prevent overfitting (References: *Genomics*. 2013, 102:237; *Bioinformatics*. 2010, 26:261).

5) ESI prediction task: Inspired by your great suggestion (Q5), we used the TransDSI deep learning framework to predict the protein ubiquitination ligase E3-substrate interaction (ESI). On an independent ESI test set, we found that this deep learning framework outperformed other machine learning systems. These results demonstrate that the TransDSI deep learning framework has certain generalization ability. We have presented all the above speculations in the section "**Discussion**" in our revised manuscript.

2. Selection process of golden negative datasets

Following your suggestion, we performed 30 iterations of random sampling for the negative set construction. Notably, the AUROC and AUPRC of TransDSI on the independent test set exhibited remarkable stability, with standard deviations of only 0.017 and 0.025, respectively (**Supplementary Table 2**). These results demonstrate that TransDSI has excellent robustness and the potential for practical applications. We have presented these results in the revised manuscript, which makes our TransDSI more convincing. Please refer to the section "**Results** – TransDSI has the ability to predict true DSIs", and **Supplementary Table 2** in the revised manuscript for further details.

Q2: While the proposed model showcases a well-structured design, it remains essential to evaluate its predictive performance in comparison to traditional machine learning (ML) methods. To establish the efficacy of DeepDSI in predicting DUB-substrate interactions, it is crucial to directly compare its performance against these established ML methods using the same dataset and features. If DeepDSI demonstrates superior performance over these conventional ML methods in such a head-to-head comparison, it would serve as a promising testament to the model's effectiveness in the realm of DUB-substrate interaction prediction. This comparative analysis would provide a clear demonstration of DeepDSI's potential and its capacity to outperform existing techniques.

A: Thanks for your great suggestion. This point was also raised by Reviewer 1(Q6).

In this revision, we constructed five additional DSI prediction systems employing machine learning methods including random forest (RF), support vector machine (SVM), eXtreme gradient boosting (XGBoost), logistic regression (LR), and K-nearest neighbors (KNN) (see section "**Results** - TransDSI has the ability to predict true DSIs"). To demonstrate the performance of the TransDSI deep transfer learning framework, we used the same training dataset and protein sequence features as that of TransDSI to construct five DSI prediction systems. When evaluated against the same independent test set, TransDSI outperformed other prediction systems, exhibiting the largest area under the ROC curve (TransDSI: AUROC=0.75; RF: AUROC=0.70; XGBoost: AUROC=0.66; SVM: AUROC=0.66; LR: AUROC=0.66).

We have added these comparison results to our revised manuscript. Please refer to the section "**Results** - TransDSI has the ability to predict true DSIs", **Fig. 2 (Figure R3-1** in this letter), **Supplementary Tables 2 and 3** in the revised manuscript for details.

Figure R3-1. Performance evaluation of TransDSI. (a-d) Performance of various models (TransDSI, UB2, and five machine learning models) for the prediction of DSIs, evaluated through 5-fold cross validation and independent test, respectively. Five machine learning models are random forest (RF), support vector machine (SVM), eXtreme gradient boosting (XGBoost), logistic regression (LR), and K-nearest neighbors (KNN). The ROC curves of the assessment models demonstrate sensitivity and

specificity (**a, b**) and the PR curves of the assessment models precision and recall (**c, d**) against a particular prediction score cutoff, with each point on the curves representing the respective values. The 95% confidence intervals (95% CIs) of the sensitivity (precision) at the given specificity (recall) points are computed. The reference line indicates a non-informative prediction with an AUROC of 0.5 (**a, b**) or a prediction with a constant F1 score across different thresholds (**c, d**). (**e**) Multidimensional association features for known DSIs, predicted DSIs, and randomly screened DRIs. The feature scores include GO term similarity in terms of biological process (GO BP), cellular component (GO CC) and PPI network-based neighborhood similarity (PPI.NS). The colored bars represent the average value of various association features, with the error bars marking 95% confidence intervals. Wilcoxon test is used to test the difference between DRIs and known or predicted DSIs. Our results show that for all kinds of features, the average score of DRIs is significantly lower than that of known and predicted DSIs (* P-value < 0.05, *** P-value < 0.001). DRI, DUB-random protein interaction. This figure is **Fig. 2** in the revised manuscript.

Q3: The authors' assertion in the Abstract that "the performance of DeepDSI exceeds state-of-the-art methods" could potentially lead to an exaggerated perception of DeepDSI's predictive prowess. It's important to note that the comparison in this work was limited to UbiBrowser 2.0, a database primarily designed for E3 ligase/DUB–substrate interactions in eukaryotic species. The DSI prediction was executed using a protocol similar to UbiBrowser 1.0, incorporating elements like Gene Ontology (GO) terms, domain pairs, DUB recognition consensus motifs, and PPI networks into a Bayesian model for predicting human DUB-substrate interactions.

Given the scarcity of experimentally validated DSIs, it's understandable why the authors opted to compare their model solely with UbiBrowser. However, to enhance the rigor of their evaluation, it is advisable for the authors to explore a broader range of machine learning methods for comparison. This should be done using an independent testing dataset. It's crucial that the authors explicitly clarify that the performance comparison was exclusively conducted with UbiBrowser 2.0 to provide accurate context to readers.

A: Thanks for your reminder.

Yes, we should not use the phrase "exceeded the SOTA method" before comparing with more methods. Our group's UbiBrowser 2.0 is the only publicly available bioinformatics tool that can predict proteome-wide DSIs (*Nucleic Acids Res.* 2022, 50: D719). Meanwhile, according to your great suggestions(**Q2**),

we have established other DSI prediction models based on five machine learning methods for comparison in this revision (see the section "**Result** - TransDSI has the ability to predict true DSIs" of the revised manuscript). The results show that our TransDSI model outperforms other algorithms in the DSI prediction task.

We have changed the related text, added the above analysis results and discussion to the revised manuscript. Please see the sections "**Abstract**" and "**Results**", **Fig. 2 (Figure R3-1** in this letter) and **Supplementary Tables 2 and 3** in the revised manuscript for details.

Q4: While the experimental validation of two pairs of predicted DSIs serves to showcase DeepDSI's effectiveness, the detection of DSI binding sites within this work presents challenges. Without annotated experimentally validated DSI binding site data, evaluating the accuracy of predicted binding sites between DUBs and substrates becomes complex. Even with the inclusion of MD simulation results for the USP7-DNMT1 interaction, the lack of annotated information for experimentally confirmed DSI binding sites impedes a comprehensive assessment of the predicted sites.

A point of confusion arises from Fig. 3, where the USP7-DNMT1 interaction is depicted without the ubiquitin component. It's crucial to include ubiquitin, which attaches to specific lysine residues and plays a pivotal role in illustrating how DUBs (like USP7) recognize ubiquitinated substrates (such as DNMT1). Regarding the significance of the 'KGK' motif as a potential binding site in the USP7-DNMT1 interaction, clarifying the location of ubiquitin (Ub) within the USP7-DNMT1 complex is important. Understanding where ubiquitination occurs can significantly inform the detection of DSIs' binding sites.

To enhance the precision of binding site detection for DSIs, it's recommended that the authors take into consideration the information about ubiquitination sites. This inclusion would contribute to a more accurate and comprehensive analysis of DSI binding interactions.

A: Thanks for your great comments

1. Prediction and assessment of DSI binding sites

As you pointed out, it is a great challenge to establish a bioinformatics system for predicting the binding sites between DUBs and substrates. After all, currently we have no sufficient experimentally confirmed DSI binding site data for training and comprehensive assessment.

In the revision, we compiled a collection of DSI binding sites documented in literature (9 sites, involving 1 DUB and 5 substrates) to evaluate our PairExplainer module. While we can predict one of the two known USP7 recognition motifs (50%) and one of the two known substrate binding domains on USP7 (50%) (**Table R3-1** in this letter and **Supplementary Table 7**), it is still far from ideal.

In **Fig. 3c** of the manuscript, we illustrate the predicted binding site of USP7-DNMT1. This figure shows the actual protein structure for the complex of USP7-DNMT1, which was determined by X-ray crystallography (Cheng *et al.*, *Nat Commun.* 2015, 6:7023), not a dynamic simulation.

Indeed, it would be very exciting if the poly ubiquitin chain could be present in the structure of DUB-substrate complex. The accurate determination of the dynamic structure of the DUB machine is crucial for illustrating how USP7 recognizes DNMT1 and removes ubiquitin. Unfortunately, Cheng *et al.* didn't provide structural information about the ubiquitin for the protein binding in their paper (*Nat Commun.* 2015, 6:7023), and we couldn't get this information elsewhere.

Thanks to your great comments, we have revised the section on DSI site prediction in the revised manuscript (section "**Results** - Explainable model of TransDSI provides partial insights into the protein structural basis of DSI").

2. Considering ubiquitin modification sites datasets for DSI prediction

In fact, we have been considering incorporating the rich experimental data on ubiquitin modification sites into DSI prediction. However, our analysis did not reveal any association between the binding sites and ubiquitination sites.

First, we compiled data on all nine DSI binding regions reported in the literature (**Table R3-1** in this letter, **Supplementary Table 7 in the revision**), including one DUB (USP7) and its five substrates DNMT1/UHRF1/MDM2/MDM4/TP53 (**Table R3-1**). We observed that these binding sites did not overlap with known ubiquitination sites (from the Ubsite database, *BMC Syst Biol.* 2016. 10: S6).

Next, we investigated the regions of 40/60/80/100 amino acid residues centered on the DUB-substrate binding sites. We found that the number of ubiquitination sites in these regions did not show a significant enrichment or depletion trend compared to those in randomly selected regions of the same substrate protein (**Figure R3-2**).

Of course, these analyses are based on the limited data available at present. In the future, as more DSI binding site data becomes available, it may be possible to find the correlation. This would allow us to fully utilize the value of existing ubiquitination modification site data.

In our manuscript, we have added discussion for all the above results and analysis. Please see the section "**Discussion**" and **Supplementary Fig. 5** for details.

Table R3-1. All experimentally validated DSI binding sites reported in the literature*.

DUB				Substrate				Source
Gene Symbol (DUB)	Substrate binding domain	Position	Identified by TransDSI	Gene Symbol (SUB)	DUB recognition motif	Position	Identified by TransDSI	
USP7	MATH domain	68-195	Yes	MDM4	P/AxxS	8-12	No	MEDLINE:20713061
USP7	MATH domain	68-195	Yes	MDM4	P/AxxS	398-402	No	MEDLINE:20713061
USP7	MATH domain	68-195	Yes	TP53	P/AxxS	359-363	No	MEDLINE:16402859
USP7	MATH domain	68-195	Yes	TP53	P/AxxS	364-368	No	MEDLINE:16474402
USP7	MATH domain	68-195	Yes	MDM2	P/AxxS	226-230	No	MEDLINE:16402859
USP7	MATH domain	68-195	Yes	MDM2	P/AxxS	147-151	No	MEDLINE:16474402
USP7	MATH domain	68-195	Yes	MDM2	P/AxxS	397-401	No	MEDLINE:20713061
USP7	ICP0-binding domain	622-801	No	DNMT1	KxxxKxK	1111-1115	Yes	MEDLINE:25960197
USP7	ICP0-binding domain	622-801	No	UHRF1	KxxxKxK	646-650	Yes	MEDLINE:26046769

* This table is **Supplementary Table 7** in the revised manuscript.

Figure R3-2. Enrichment analysis of ubiquitination sites within all known DSI binding regions. All nine DSI binding regions were compiled from literatures, including one DUB (USP7) and its five substrates DNMT1/UHRF1/MDM2/MDM4/TP53 (**Supplementary Table 7**). We investigated the regions of 40/60/80/100 amino acid residues centered on the DUB-substrate binding region. We found that the number of ubiquitination sites in this region did not show a significant enrichment or depletion trend compared to those in randomly selected regions from the same substrate protein. Scatter plots depict the \log_2 ratio of the number of ubiquitination sites around known DUB substrate binding regions versus random regions, across varying window sizes (**a**: 40, **b**: 60, **c**: 80, **d**: 100). (Red: more ubiquitination sites around binding region than random; blue: fewer ubiquitination sites around binding region than random). N_{site} : number of ubiquitination sites around known substrate binding region; N_{random} : number of ubiquitination sites around random region. Gray dashed reference line: $N_{site} = N_{random}$. Red asterisk: no ubiquitination sites in the binding region ($N_{site}=0$). This figure is **Supplementary Fig. 5** in the manuscript.

Q5: The data statistics within UbiBrowser 2.0 indicate a substantial disparity between the numbers of E3-substrate interactions (ESIs) (4,068) and DUB-substrate interactions (DSIs) (967). In light of this, it's highly advisable for the authors to apply a similar framework to the prediction of ESIs using the proposed DeepDSI model. The abundant and balanced dataset of ESIs can provide a robust foundation for developing and evaluating a deep learning-based model.

By adopting the same framework for ESIs, the authors would have the opportunity to comprehensively assess the effectiveness and capabilities of the proposed DeepDSI model. Since it performed well on DSI prediction, its application to ESIs should indeed yield promising results. The considerable dataset size for ESIs ensures a solid training foundation for the deep learning model, potentially bolstering its predictive performance.

It's important to emphasize that having substantial datasets for both model training and performance testing is paramount. This approach will provide more conclusive evidence regarding the efficacy of the proposed DeepDSI model. Expanding the model's evaluation to ESIs can enhance its overall credibility and applicability.

A: Thanks for your excellent suggestion.

Following your suggestion, we constructed a novel model (TransESI) for predicting ESIs following the same protocol of TransDSI framework. We performed model training and performance evaluation for TransESI as follows.

First, we constructed the gold standard positive/negative dataset of ESIs using a similar approach to that for DSIs. Specifically, we collected all literature abstracts that mention potential E3-substrate interactions from PubMed. Then, a panel of three experts was established to manually curate these publications. Only ESIs that met the following criteria were included in our GSP: 1) The text of the publication explicitly states that the E3 ubiquitinates the substrate; 2) The publication provides evidence that the E3-substrate interaction is functional. We constructed the negative set by randomly selecting an equal number of nodes from the complement graph of the known ESI network, ensuring identical E3 connectivity distributions between the negative and positive sets. All interactions within the negative set are derived from PPI data in the BioGRID database (Released 25 January 2022). Importantly, none of the interactions in the GSN dataset are present in the GSP dataset.

Next, we split the gold standard ESI dataset into a training set and an independent test set. The training set comprised 2367 ESIs identified before June 2018, while the independent test set comprised

472 ESIs identified from June 2018 to August 2021.

Then, we trained the model on ESI data following the same protocol of TransDSI framework and evaluated it on the independent dataset.

Notably, the TransDSI deep learning framework also demonstrates satisfactory performance on ESI prediction with an AUROC of 0.70, outperforming other machine learning methods (including random forest, support vector machine, eXtreme gradient boosting, logistic regression, and K-nearest neighbors (**Figure R3-4**).

These added results of ESI prediction provide more conclusive evidence regarding the efficacy of the proposed TransESI (TransDSI) model, enhancing its overall credibility and applicability.

Please refer to Section "**Methods** - Construction of TransESI model for predicting ubiquitin E3 ligase-substrate interactions", **Figure R3-3, R3-4** in this letter (**Supplementary Fig. 2, 3** in the revised manuscript) for details.

Figure R3-3. Framework of TransESI model. TransESI has the same framework as TransDSI in **Fig. 1 of revised manuscript**. The model comprises of three components: **(a) Protein coding**: This module takes the primary structures of human proteins including E3s (P_{E3}), substrates (P_{SUB}) and other proteins (P_1 - P_6) as input and generates a variety of features for use in downstream deep learning modules. The amino acid sequences of proteins are encoded using the CT method and serve as protein sequence features X (darker colors indicate higher conjoint triad frequency). Additionally, an SSN was created using BLAST and transformed into a normalized sequence similarity matrix A (darker colors indicate higher similarity). **(b) Self-supervised module for embedding protein sequence features**: This module employs a VGAE consisting of a GCN encoder and a dot product decoder, which is used to generate a

pre-trained encoder based on the evolutionary information from the SSN and the protein sequence features (see **Methods** for details of σ, μ and z). (c) Semi-supervised module for predicting ESI (ESI-Predictor): The GCN encoder of ESI-Predictor is initialized using the parameters transferred from the self-supervised module to produce the protein embeddings. The embeddings of E3s and their corresponding substrates involved in GSD are concatenated and utilized for fine-tuning the semi-supervised module. The final prediction score ("TransESI Score") for each candidate ESI is obtained by feeding the concatenated embeddings to an MLP. This figure is **Supplementary Fig.2** in the revised manuscript.

Figure R3-4. Evaluation of the performance of TransESI. We constructed and assessed TransESI models following the same protocol as TransDSI. TransDSI deep learning framework demonstrates satisfactory performance on ESI prediction with the TransESI model achieving an AUROC of 0.70, surpassing other machine learning methods in predictive efficiency. (a, b) Performance of various models (TransESI and five machine learning models) for the prediction of ESIs, evaluated through independent test. Five machine learning models are random forest (RF), support vector machine (SVM), eXtreme gradient boosting (XGBoost), logistic regression (LR), and K-nearest neighbors (KNN). The ROC curves of the assessment models demonstrate sensitivity and specificity (a) and the PR curves of the assessment models precision and recall (b) against a particular prediction score cutoff, with each point on the curves representing the respective values. The 95% confidence intervals (95% CIs) of the sensitivity (precision) at the given specificity (recall) points are computed. The reference line indicates a non-informative prediction with an AUROC of 0.5 (a) or a prediction with a constant F1 score across different thresholds (b). This figure is **Supplementary Fig.3** in the revised manuscript.

Reviewer #1 (Remarks to the Author):

The authors have extensively and comprehensively addressed all the concerns raised by the reviewers. The revision significantly improved their work, although some minor issues remain to be addressed.

Q1: There are few statements needing a re-consideration for a better declaration, as listed below. Lines 27-30: "Two predicted DUBs (USP11 and USP20) and for FOXP3, along with two predicted substrates (AR and TP53) for USP22, ..., contributing to tumor immune escape-related drug target discovery and precise application of anti-tumor agent, respectively". I would argue that it may be too early to declare USP11 and USP20 as the potential drug targets, as there is no direct functional data is provided in the manuscript to demonstrate their involvement in tumor immune evasion. Lines 469-471: 'Our results demonstrate that UCHL1 promotes tumor progression in HCC by stabilizing PKM2 via deubiquitination, thereby providing new insights into the molecular mechanism of HCC pathogenesis'. Similarly, this statement is a bit too strong as these results were purely based on association analysis with clinical parameters, which provided important indications but still needs further functional investigation. Lines 71-72: 'UbiBrowser 2.0 remains the only publicly available bioinformatics tool capable of proteome-wide DSIs prediction' and lines 148-149: 'which is the only publicly available bioinformatics tool that can predict proteome-wide DSI'. There were few published tools available, although not as comprehensive as this work.

Q2: Why does Myc-USP22 Western Blotting have two adjacent bands in the immunoprecipitates (IP-flag) while not in whole cell (WCL), as shown in Figures 5b, 5c, 5f?

Reviewer #1 (Remarks on code availability):

A README file with enough instructions for installing and running the application was provided.

Reviewer #4 (Remarks to the Author):

This study developed a novel and significant computational algorithm to predict the DUB-substrate interactions, which has a large impact on drug development. I think the authors have done a great job in addressing the comments raised by the reviewers. Through this revision, the authors further demonstrated the value of their approach by incorporating experimental validation, side-by-side comparisons with conventional machine learning methods, additional ESI prediction, and real structural insights. This is a great work and would be highly valuable to a broad biomedical community.

Reviewer #4 (Remarks on code availability):

The code is clearly documented.

The reviewers' comments verbatim are in **blue**. Our responses are in plain black text. Our corresponding changes were highlighted in **red** in the manuscript.

Response to Reviewer 1:

Comments for the Author:

The authors have extensively and comprehensively addressed all the concerns raised by the reviewers. The revision significantly improved their work, although some minor issues remain to be addressed.

A: Thanks for your comments.

Your comments have greatly improved the manuscript. And we have also made revision according to your great comments pointed out this time.

Q1: There are few statements needing a re-consideration for a better declaration, as listed below. Lines 27-30: "Two predicted DUBs (USP11 and USP20) and for FOXP3, along with two predicted substrates (AR and TP53) for USP22,, contributing to tumor immune escape-related drug target discovery and precise application of anti-tumor agent, respectively". I would argue that it may be too early to declare USP11 and USP20 as the potential drug targets, as there is no direct functional data is provided in the manuscript to demonstrate their involvement in tumor immune evasion.

A: Thanks for your comment. This is the revised "**Lines 27 -30**".

'Two predicted DUBs (USP11 and USP20) for FOXP3 are validated by "wet lab" experiments, along with two predicted substrates (AR and p53) for USP22.'

Q2: Lines 469-471: 'Our results demonstrate that UCHL1 promotes tumor progression in HCC by stabilizing PKM2 via deubiquitination, thereby providing new insights into the molecular mechanism of HCC pathogenesis'. Similarly, this statement is a bit too strong as these results were purely based on association analysis with clinical parameters, which provided important indications but still needs further functional investigation.

A: Thanks for your comments. This is the revised "**Lines 469-471**".

'Our results suggest that UCHL1 may contribute to tumor progression in HCC by stabilizing PKM via deubiquitination, thereby providing **important indications into the potential molecular mechanism of HCC pathogenesis.**'

Q3: Lines 71-72: 'UbiBrowser 2.0 remains the only publicly available bioinformatics tool capable of proteome-wide DSIs prediction' and lines 148-149: 'which is the only publicly available bioinformatics tool that can predict proteome-wide DSI'. There were few published tools available, although not as comprehensive as this work.

A: Thanks for your comments.

This is the revised **Lines 71-72:** 'UbiBrowser 2.0 is a popular publicly available bioinformatics tool for proteome-wide DSI prediction, representing a significant advancement in the field.'

Revised Lines 148-149: 'which is a popular publicly available bioinformatics tool for predicting proteome-wide DSIs.'

Q4: Why does Myc-USP22 Western Blotting have two adjacent bands in the immunoprecipitates (IP-flag) while not in whole cell (WCL), as shown in Figures 5b, 5c, 5f?

A: Thank you for your insightful inquiry.

As you mentioned, in the immunoprecipitates (IP-flag) but not in the WCL, Myc-USP22 Western Blotting have two adjacent bands. The presence of heavy chain of antibody in the IP sample causes this phenomenon.

Co-IP is a common approach to study protein-protein interactions that uses an antibody to immunoprecipitate the antigen (bait protein) and co-immunoprecipitate any interacting proteins (prey proteins). Co-IP methods that use Protein A or G result in co-elution of the antibody heavy and light chains that may co-migrate with the target bands. As shown in the figure below (which is Figure 7c of *Nat Cell Biol.* 2015, 17: 1169), the authors also observed two adjacent bands in immunoprecipitates, similar to our Co-IP results.

To avoid the possible confusion, we have made the revision in Figure 5, and have also marked this band as IgG HC (similar to Figure 7c of *Nat Cell Biol.* 2015, 17: 1169).

Revised Figure 5:

Response to Reviewer 2:

No more comments.

A: Thanks for your great comments.

Response to Reviewer 4:

Comments for the Author:

This study developed a novel and significant computational algorithm to predict the DUB-substrate interactions, which has a large impact on drug development. I think the authors have done a great job in addressing the comments raised by the reviewers. Through this revision, the authors further demonstrated the value of their approach by incorporating experimental validation, side-by-side comparisons with conventional machine learning methods, additional ESI prediction, and real structural insights. This is a great work and would be highly valuable to a broad biomedical community.

A: Thanks for your high evaluation.

We hope this will be a highly valuable resource for the broad biomedical community.